# ADVANCING BEYOND IDENTIFICATION: MULTI-BIT WATERMARK FOR LARGE LANGUAGE MODELS

## ABSTRACT

We propose a method to tackle misuses of large language models beyond the identification of machine-generated text. While existing methods focus on detection, some malicious misuses demand tracing the adversary user for counteracting them. To address this, we propose "Multi-bit Watermark via Position Allocation", embedding traceable multi-bit information during language model generation. Leveraging the benefits of zero-bit watermarking (Kirchenbauer et al., 2023a), our method enables robust extraction of the watermark without any model access, embedding and extraction of long messages ($\geq$ 32-bit) without finetuning, and maintaining text quality, while allowing zero-bit detection all at the same time. Moreover, our watermark is relatively robust under strong attacks like interleaving human texts and paraphrasing.

## 1  INTRODUCTION

How can we take a step further from merely identifying machine-generated text to proactively tackling misuses of large language models? The emergence of human-like language models has necessitated the development of various methods to detect machine-generated texts through techniques such as zero-shot detection, supervised training, watermarking, and more (Mitchell et al., 2023; Wang et al., 2023b; Kirchenbauer et al., 2023a; Krishna et al., 2023). These endeavors focus on the crucial task of *identifying* machine-generated content, which serves as a pivotal step in mitigating the potential harm caused by such text.

However, when it comes to more pernicious misuses of large language models, such as the dissemination of misinformation and propaganda on social media platforms, the stakes are considerably higher, potentially leading to the erosion of social trust (Valenzuela et al., 2022). Notable instances that exploited automated bots in the past include manipulating an election campaign (Badawy et al., 2018), spreading disinformation about the Russian invasion of Ukraine (Pierri et al., 2023), and promoting products through fake reviews (Annie, 2023). With the rapid pace at which large language models are currently developed, similar threats will be automated in a much more rapid and delicate manner in the future.

In these scenarios, merely identifying the machine-generated text may not suffice for the language model providers. Instead, the ability to trace back to the adversary user responsible for generating the content becomes pivotal in counteracting such misuses. By doing so, the API providers can take a precursory measure to ban these users from their systems. More importantly, this allows media and social platforms, along with API providers, to collaborate with law enforcement authorities and take more decisive actions. All in all, watermarking the user information (or part thereof) can hold the adversary user accountable for potential harms facilitated through language model APIs without having to store user queries (Krishna et al., 2023), which would be prohibitively expensive and concern ordinary users who value privacy. Additionally, watermarking can enable language model providers to bind meta-data (e.g. model versions) for tracing the provenance of the language model output.

All this can be achieved by embedding multi-bit information. In this study, we demonstrate that this can be realized on top of the recently proposed zero-bit watermarking method (Kirchenbauer et al., 2023a) in an extremely simple way without sacrificing the text quality. Zero-bit watermarking

works by randomly favoring a "greenlist" of tokens at each generation. The selection of a greenlist from the vocabulary set is determined by a random seed generated by a pseudo-random function.

Our proposed method called **M**ulti-bit watermark via **P**osition **A**llo**c**ation (MPAC) first allocates each token pseudo-randomly onto a single position of the message to be embedded. Then the message content at the allocated position decides which subset of tokens to favor. We show that the zero bit watermarking method can be viewed as a special case of encoding the same single bit message. To increase load capacity, we can further partition the vocabulary into multiple "colored" lists instead of a single green list, effectively encoding multiple states for every token. By viewing our method through coding theory, we analyze what factors affect the performance of our method and devise techniques for improvement. We also discuss and analyze the limitations of MPAC in Section 5 – namely, the trade-off between watermark detection and bit-width.

Since our method works on top of zero-bit watermarking, it leverages most of the advantages: (1) Multi-bit message can be extracted without access to the model parameters or the API, allowing other parties to extract the adversary information (e.g. timestamp, ID) if given access to the extraction algorithm. (2) It can be done on the fly without pretraining or finetuning the model and can embed and extract long messages ($\geq$ 32-bit) with negligible overhead. (3) The watermark is not fragile against realistic corruptions such as interleaving with human texts or paraphrasing. This has not been previously demonstrated in other post-processing multi-bit watermarks (Yoo et al., 2023) or stenography methods (Ziegler et al., 2019; de Witt et al., 2023). (4) Finally, our watermarking framework can distinguish between machine and human text and simultaneously embed multi-bit information while maintaining the same text quality as its zero-bit counterpart. Our experiments demonstrate that 8-bit messages can be embedded effectively in short text lengths ($\sim$100 tokens) with over 90% bit accuracy. We hope this opens up new research directions to proactively counteracting malicious use cases of language model APIs.[1]

## 2 RELATED WORKS

Watermarking has been studied in various types of multimedia such as image (Potdar et al., 2005), video (Asikuzzaman & Pickering, 2017), audio (Hua et al., 2016), and natural language (Topkara et al., 2005). Following previous works (Zhu et al., 2018; Luo et al., 2020), we use the term watermarking to denote embedding information into natural language in a manner that is robust against possible attacks given a watermarked text – in our case, this is the output generated by a language model given the prompt. This differs from steganography(Cheddad et al., 2010; Fang et al., 2017; Ziegler et al., 2019; de Witt et al., 2023), which focuses more on the undetectability of a secret message that is embedded in the multimedia rather than robustness. For instance, Ziegler et al. (2019) rely on arithmetic coding every token, rendering a burst error when the first token is substituted or removed.

Recently, methods relying on neural networks have shown progress in natural language watermarking, outperforming traditional methods that rely on rule-based watermarks (Topkara et al., 2006b;a; Atallah et al., 2001). Abdelnabi & Fritz (2021) proposed an end-to-end framework where a decoder network predicts the encoded message. Yang et al. (2022) improved upon the quality of the watermarked text by using an algorithmic approach. Yoo et al. (2023) focused on robustness and capacity, outperforming previous works on the two aspects. However, since the proposed method works at sentence-level, any addition or removal of a sentence will fail to extract the watermark. Moreover, these works cannot distinguish non-watermarked texts, making them unsuitable for distinguishing between machine text and human text.

Meanwhile, directly watermarking language models in a zero-bit manner during token generation has emerged as a promising approach for distinguishing language model outputs from human text (Kirchenbauer et al., 2023a; Aaronson & Kirchner, 2023) while achieving robustness against realistic attacks as it can reinforce the watermark every token (Kirchenbauer et al., 2023b). Several works have improved upon Kirchenbauer et al. (2023a), e.g., in low entropy generation tasks such as code generation (Lee et al., 2023), undetectability of the watermark (Christ et al., 2023), and its robustness (Munyer & Zhong, 2023). We focus on extending the prior work for a more proactive counteraction towards identifying malicious users of language models by embedding *any* information while maintaining the key advantages.

---

[1]https://github.com/anoymous92874838/multibit-watermark-for-llms

Concurrent to our work, Fernandez et al. (2023a) propose a technique for encoding a multi-bit message by providing a message-specific greenlist through shifting the vocabulary list dependent on the message. Similarly, Wang et al. (2023a) use the message content as the hashing key before selecting the greenlist and further utilizes an auxiliary language model for enhancing text quality. Crucially, both works use the entire message content directly during embedding as input to the random seed generator, which requires computing through the exponential number of possible messages during decoding. This restricts the length of the message due to computational and/or memory limitations. To give a rough estimate of the required message length for encoding a user ID, consider the POSIX (Group, 2018) standard used when creating usernames in operating systems. 65 characters ($\sim$7 bits) are permitted by POSIX, meaning at least 35 bits are required to encode a username of 5 characters. Accordingly, works in image watermarking embeds messages easily over 32-bits (Zhu et al., 2018; Zhao et al., 2023; Fernandez et al., 2023b). Our method allows the embedding of long messages without any added latency by encoding each bit position independently.

## 3 METHOD

We briefly review zero-bit watermarking introduced by Kirchenbauer et al. (2023a) and elaborate on extending this method to multi-bit watermarking. Then, we analyze our framework from the lens of coding theory and introduce additional techniques for improving the watermark performance.

### 3.1 ZERO-BIT WATERMARKING (KIRCHENBAUER ET AL., 2023A)

A watermark is embedded by biasing the language model to output a certain subset of tokens. Given an autoregressive language model that predicts the next token with vocabulary $V$, a subset of tokens is randomly selected from the vocabulary at each token step $t$ and forms a green list $\mathcal{G}_t$. The logit scores $l_t \in \mathbb{R}^{|V|}$ are modified towards selecting the green-listed tokens in favor of the other tokens by adding a bias term $\delta$ to the logits in $\mathcal{G}_t$. Instead of fixing the greenlist using rule-based heuristics such as spelling or synonym variations (He et al., 2022), the greenlist is selected (pseudo-)randomly at each time step to minimize a noticeable shift in text distributions. At each time step, a seed $s$ is outputted depending on the previous $h$ tokens using a pseudo-random function $f : \mathbb{N}^h \to \mathbb{N}$, and $s$ is used to sample $\mathcal{G}_t$ from $V$.

At decoding, the greenlist can be recovered by using the same pesudo-random function $f$. The presence of a watermark is determined by counting the number of tokens in the greenlist. For a human-generated text that has no knowledge of the greenlist rule, a token will be from the greenlist with the probability $\gamma$, the proportion of the greenlist size compared to the entire vocabulary. Without the knowledge of the greenlist (null hypothesis), the number of tokens in the greenlist ($g$) follows a binomial distribution. (Kirchenbauer et al., 2023a) used the normal approximation to the binomial distribution to compute the $z$-statistics for a text with $T$ tokens: $z = \frac{g - \gamma T}{\sqrt{\gamma(1-\gamma)T}}$.

### 3.2 MPAC: EXTENDING TO MULTI-BIT WATERMARK

We first present an overview of our method and further elaborate on the details in the subsequent section. The objective of multi-bit watermarking is to embed and extract a message $\mathbf{m} \in \Sigma^b$ where $\Sigma$ denotes the $r$-nary possible strings, or more commonly referred to as the alphabet. For a binary message, $\Sigma = \{0, 1\}$. We let $p \in \{0, \dots, b-1\}$ denote the position of the message and $\mathbf{m}[p] \in \{0, \dots, r-1\}$ the message content at that position. Hereafter, we use $[a]$ to denote the integer set $\{0, \dots, a-1\}$.

Our proposed method **M**ulti-bit watermarking via **P**osition **A**llo**c**ation (MPAC) works by partitioning the tokens to message positions and enlarging the size of the alphabet through color-listing. First, notice that zero-bit watermarking can be viewed as watermarking a single bit of information stating the existence of a watermark ($\mathbf{m}$=0). In essence, each token generated by the language model is a signal in favor of the watermark (See Fig. 1-Right).

**Message Encoding** In MPAC, we allocate the signals (tokens) into multiple positions. For instance, when the message content at a position is '0', we sample from the greenlist, while doing otherwise when the message is '1'. This allows encoding multi-bit messages of arbitrary length as long as the language model generates sufficient tokens. To further increase the bit capacity, we color the

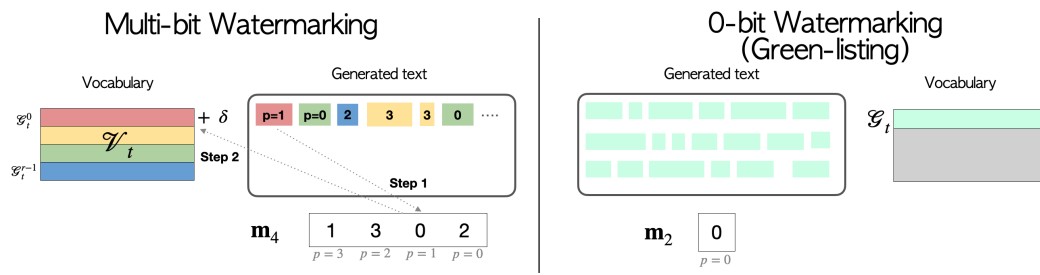

Figure 1: An overview of our method MPAC. The number inside a token (e.g. $p = 1$) denotes the allocated position, while the color signifies the message content at that position. At Step 1, a position is sampled prior to generating a token. At Step 2, the message at that position determines the token subsets to favor. *Right:* Zero-bit watermarking can be viewed as a special case of multi-bit watermarking.

vocabulary set with $r$ "colors" instead of using a single greenlist, increasing the alphabet to $\Sigma = [r]$. The number of colors can be determined by the greenlist proportion $\gamma$, i.e. $r = \lfloor \frac{1}{\gamma} \rfloor$. Thus, this allows encoding $r$ states into each token as opposed to encoding a binary state (whether the token is selected from the greenlist or not). Given a binary message of length $b$, the message is convereted to radix $r$ attaining $\mathbf{m}_r \in [r]^{\tilde{b}}$ where $\tilde{b} = \lceil \frac{b}{\log_2 r} \rceil$. In Figure 1 Left, we illustrate the case of $r = 4$ and $b = 8$, where the 8-bit message is converted into radix 4, resulting in an effective message length of 4 ($\tilde{b} = 4$)[2]. At each token generation, the message content at the assigned position $p$ determines which colorlist to add $\delta$ to. If the message content is '0', the tokens from the first list (red in Fig. 1) are favored. We discuss the design choices for allocating the positions in the next section.

Our method is extremely easy to implement over the zero-bit watermarking scheme. We highlight the steps in colors that are specific to ours. Given $t-1$ prefix tokens $X_{1:t-1}$, bit message $\mathbf{m}$, its r-radix converted form $\mathbf{m}_r$, and pseudo-random function $f$, the $t^{\text{th}}$ token is generated by

1. Compute hash of tokens $s = f(X_{t-h:t-1})$. Use $s$ as a seed for a random number generator.
2. $p \leftarrow \texttt{sample}([b])$        # $p$ is the position of the message.
3. $m \leftarrow \mathbf{m}_r[p]$              # $m$ is the message content at position $p$.
4. Permute vocabulary $\mathcal{V}_t$ using $s$ as seed.
5. Partition $\mathcal{V}_t = [\mathcal{C}_t^0, \cdots, \mathcal{C}_t^{r-1}]$ discarding remainders if any.
6. Add $\delta$ to token logits in $\mathcal{C}_t^m$.

Note that zero-bit watermarking can be seen as a special case of embedding the same single bit message ($b = 1$, $r = 2$, and $\mathbf{m} = 0$) as shown in Figure 1-Right.

**Message Decoding** Given a watermarked language model output, we determine the position and which colorlist each token is from and increment the number of tokens in the colored lists. For instance, for the $t^{\text{th}}$ token with message position $p = i$ and the $j^{\text{th}}$ colorlist $\mathcal{C}_t^j$, we increment the counter $\mathbf{W}_i[j]$. After computing this on the entire text segment, we predict the message content by taking the colorlist with the most tokens for each position. A more detailed algorithm is shown in Algorithm 1.

MPAC encodes and decodes each bit position of the message independently, which brings a negligible increase in the computation as the message length is increased. This is in contrast with recent works (Wang et al., 2023a; Fernandez et al., 2023a)[3] that have to compute the likelihood of all the possible messages during decoding, which makes the embedding of long-length messages infeasible due to exponentially growing ($\mathcal{O}(2^b)$) computations during decoding without additional techniques to avoid this. In contrast, our algorithm can encode 64-bit messages ($2^{64}$ messages) as will be shown in the subsequent section.

---

[2]Hereafter, we use $b$ instead of $\tilde{b}$ to denote the effective message length (dimension of $\mathbf{m}_r$).
[3]See Section 7.5 of Wang et al., 2023a

---

**Algorithm 1:** Message Decoding

---

**Input:** Watermarked text $X_{1:T}$, hash context width $h$, effective message length $\tilde{b}$
**Output:** Predicted message $\hat{\mathbf{m}}$, number of colorlisted tokens $w$

/* Initialize counter */
1   $\mathbf{W}_p[m] = 0 \; \forall p, m$
    /* Count tokens in colorlists */
2   **for** $t$ in $[h+1, T]$ **do**
3      $s = f(X_{t-h:t-1})$
4      $p = \texttt{sample}([\tilde{b}])$
5      **for** $m$ in $[r]$ **do**
6        Permute $\mathcal{V}_t$ using $s$ as seed
7        **if** $X_t \in \mathcal{G}_t^m$ **then**
8          $\mathbf{W}_p[m] \mathrel{+}= 1$

/* Predict message */
9   $\hat{\mathbf{m}}_r = $ " "
10   $w = 0$
11   **for** $p$ in $[\tilde{b}]$ **do**
12      $w \mathrel{+}= \max(\mathbf{W}_p[m])$
13      $\hat{m} = \arg\max_m (\mathbf{W}_p[m])$
14      $\hat{\mathbf{m}}_r \mathrel{+}= \texttt{str}(\hat{m})$
15   Get bit message $\hat{\mathbf{m}}$ by converting $\hat{\mathbf{m}}_r$
16   **return** $\hat{\mathbf{m}}, w$

---

### 3.3 DESIGN CHOICES THROUGH THE LENS OF CODING THEORY

Having set out the encoding and decoding algorithm of MPAC, we elaborate on the design choices and analyze what factors affect the performance using basic notions from coding theory adapted from Cover (1999):

- Encoding function is a function $E : \mathcal{M} \to \mathcal{X}$ that maps the original message into longer, usually redundant string where $\mathcal{M} \subseteq [r]^b, \mathcal{X} \subseteq \Sigma^T$. The rate of $E$ is given by $\frac{b}{T} \log_2 r$ bits/symbol.

- $p(y|x)$ is a noisy channel that models the transmission of the encoded message.

- A channel's capacity is the upper bound of the rate of an encoding function in order for a reliable transmission.

- Decoding function is a function $D : \mathcal{Y} \to \mathcal{M}$ that recovers the original message from $y$.

We first simplify our setting to embedding a single-digit message ($b = 1$), which does not lead to a loss of generality as MPAC encodes each position independently. As noted earlier, each token of a language model is a signal for embedding the message ($m$) by repetitively sampling from the $m^{\text{th}}$ colorlist. Therefore, in MPAC our **encoding** function is a repetition code that maps a redundant message content $T$ (number of tokens) times. Our **channel** is the generation process of the language model, which stochastically transmits the encoded message by sampling from the vocabulary distribution that has been modified to favor the selected colorlist. The success probability of each transmission depends on the magnitude of the bias $\delta$, the entropy of the vocabulary distribution, and, more holistically, the text distribution. The **decoding** function is the rule set out in Sec. §3.2, whereby the argmax of the colorlist is predicted as the message content, i.e. majority voting.

**Position Allocating** Coming back to our original multi-bit message setting, the rate $\frac{b}{T} \log_2 r$ signifies that having more tokens leads to increased strength of the signal (i.e. lower rate) as more tokens are assigned to each position. Ideally, allocating tokens to each position should (1) equally favor the positions and (2) be robust to potential corruptions in the watermarked text. The first criterion can be easily achieved through a rule-based scheme such as sequentially allocating positions for every $k$ tokens or deterministically cycling through the positions. However, while these schemes may effectively retain the positions when the generated text is untouched, even a single insertion or deletion of a word will lead to burst errors during decoding. This makes them extremely fragile under realistic use cases where users paraphrase or edit the generated texts.

To remedy this, we use the hashing scheme that was used for permuting the colorlists: At each token step, we sample $p \in [b]$ seeded with $s$ that was generated from the pseudo-random function $f$. This allows enjoying the relative robustness of the hashing scheme towards attacks that alter the total length (e.g. paraphrasing) or mixing snippets of human text onto the watermarked text. This is illustrated in Appendix Fig. 6a: when 20% of human texts are mixed into the watermarked texts, rule-based position allocation (cycle) almost falls to near-random chance (61%, 55%, 54%, and 51% across the bit-widths), while sampling positions via hashing maintains the watermark.

At this point, one may assume that the ratio between the number of tokens and bit-width ($\frac{T}{b}$) determines the channel's capacity. We show in Appendix Fig. 6b that this is not necessarily the case: The bit error rate increases as longer messages are embedded despite the same bits per token (BPT). This can be explained by the well-known theorem in channel coding that the noise of the signal (Shannon, 1948) also affects the channel capacity. Signal-to-noise ($SNR = \frac{\mu}{\sigma}$) precisely measures this quantity where we define $\mu$ as the mean number of tokens per position and $\sigma$ as the standard deviation. Modeling this as a uniform multinomial distribution $P \sim$ Multinomial$(T, [\frac{1}{b} \cdots \frac{1}{b}])$ for simplicity, we can get a rough estimate of SNR. Increasing both $b$ and $T$ at the same rate maintains $\mu = T/b$, but increases $\sigma = \sqrt{T(b-1)}/b$. Fig. 6b displays the theoretical SNR (= $\sqrt{T/(b-1)}$, which better explains the empirical bit error rate.

**List Decoding** List decoding is a well-established field in coding theory that decodes a list of messages that are within a certain hamming distance (Elias, 1991; Guruswami & Rudra, 2008; Guruswami, 2004). Inspired by this, we alter our decoding function to output candidate messages sorted by the level of confidence. Denoting the predicted message for position $i$ by $\hat{m}$, and the observed number of tokens in the colored list (strength of the watermark) by $w = \mathbf{W}_i[\hat{m}]$, the confidence of $\hat{m}$ should be higher if $w$ deviates from the expected mean under the null hypothesis that all colored lists are equally likely to be sampled. We define confidence at position $i$ as $c_i \propto \Pr(W_i^{\max} \leq w | H_0)$ where $W_i^{\max}$ is the maximum cell value of $W_i \overset{H_0}{\sim}$ Multinomial$(T_i, [\gamma \cdots \gamma])$ where $T_i$ is the number of tokens assigned to position $i$. The distribution of $W_i^{\max}$ is approximated using techniques from Levin (1981) (See Appendix A.9).

Our algorithm can be parameterized by the confidence bound on each position:

- Input: Best prediction $\hat{\mathbf{m}}$ found by majority voting via Alg. 1, confidence bound $c_0$
- Output: $\hat{\mathbf{m}}_1, \cdots, \hat{\mathbf{m}}_{|\mathbb{L}|} \in \mathbb{L}$ whose predictions are altered on positions with confidence under $c_0$

Empirically, we determine $c_0$ by constraining $|\mathbb{L}|$. Note that since $\hat{\mathbf{m}}$ is always the most confident message, we comprise $\mathbb{L}$ with the next confident messages. To do this, we greedily alter the positions with the lowest confidence to the colorlist with the second largest number of tokens. Note that this list decoding technique is not unique to ours and can be applied to other methods as long as the decoding stage is computationally feasible.

We believe the simplicity of our multi-bit watermark scheme via position allocation makes it generalizable to other zero-bit watermark approaches. An example is provided in Appendix A.10.

### 3.4 DETECTING MACHINE TEXT

While we can use MPAC to decode the multi-bit watermark in conjunction with another detection mechanism, MPAC alone can detect human text from watermarked text just like zero-bit watermarking. The strength of our watermark can be determined by taking the maximum cell frequency of each position, which is modeled by the confidence $c_i$ at each position. However, we found that simply modeling the number of tokens in the argmax colorlist of position $i$ as a random variable $C_i \overset{H_0}{\sim}$ Binomial$(T_i, \gamma)$ led to slightly better results where $T_i$ is the number of tokens assigned to position $i$. As $C_0, \ldots, C_{b-1}$ are independent for a fixed set of trials $(T_i, \ldots, T_{b-1})$ and have the same success probability parameter, the sum of these is a binomial random variable as well:

$$C = C_0 + \cdots + C_{b-1} \overset{H_0}{\sim} \text{Binomial}(T, \gamma) \tag{1}$$

where $T = T_0 + \cdots + T_{b-1}$. This reduces to the same random variable used in zero-bit watermarking and we can compute the z-statistics from §3.1. More discussion regarding other possible statistics is outlined in the last section of Appendix A.8. Computing $C$ is shown in Line 12 of Algo. 1.

## 4 EXPERIMENTS

### 4.1 EXPERIMENTAL SETTINGS

We use LLaMA-2-7B (Touvron et al., 2023) to generate sequences on the newslike subset of the Colossal Common Crawl Cleaned corpus (C4) dataset (Raffel et al., 2020) as the prompt for our main experiments following previous work (Kirchenbauer et al., 2023a). For watermarking and

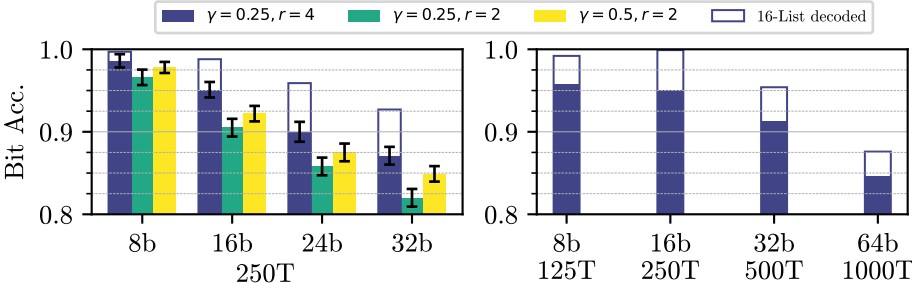

Figure 3: Clean bit accuracy with 3 standard errors for a fixed number of tokens (left) and fixed BPT (right).

generation, we follow the configurations used in Kirchenbauer et al. (2023b) unless otherwise denoted: bias $\delta = 2.0$, greenlist ratio $\gamma = 0.25$, which have shown a good trade-off between the detection performance and generation quality. Since $\gamma = 0.25$, the number of colors $r$ is 4. We embed a random $b$-bit message onto >500 samples and report the mean metrics across samples. When using the term 'bit' or 'bit-width', this denotes the initial message length and the effective message length is determined by $r$. When necessary, we also show the three standard error ranges. For list decoding, we compute a list size of 16 (in addition to the best prediction) unless otherwise noted, which corresponds to $6e^{-2}$, $2e^{-4}$, and $4e^{-9}$ of the output space for 8-bit, 16-bit, and 32-bit, respectively. More details are in Appendix A.2.

**Metrics** To measure the performance of multi-bit watermarking, we use bit accuracy following previous works in the literature (Zhu et al., 2018; Luo et al., 2020; Yang et al., 2022; Yoo et al., 2023) to measure how much of the embedded bits can be extracted without error. To compute the performance of list decoding, we take the closest message out of the candidates. For zero-bit watermark detection (i.e. machine-text detection), we use area under the ROC curve (AUROC) and the true positive rate (TPR) at various thresholds. For text quality, we use the automatic metrics used in Kirchenbauer et al. (2023b) such as perplexity (PPL) using a larger oracle model (LLaMA-2-13B) and semantic similarity based on a paraphraser model (Wieting et al., 2022, P-SP). We further discuss the validity of the metrics in Appendix A.7.

## 4.2 RESULTS

We visualize the results as graphs in the main paper. Tables are in Appendix A.4.

**Text quality is not affected by bit-width**. MPAC extends zero-bit watermarking by allocating tokens to message positions and partitioning vocabularies, which would otherwise be allocated to a single position and a single vocabulary partition. Consequently, given the same $\delta$ and $\gamma$, it only alters the text distribution to an extent that zero-bit watermarking does regardless of the bit-width. Indeed, our empirical results in Fig. 2 demonstrate that the text quality is statistically indistinguishable across bit-widths. We also show that the encoding latency, which directly experiences user experience, does not increase with bit-width. Three standard error ranges are shown.

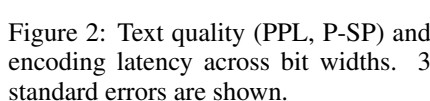

Figure 2: Text quality (PPL, P-SP) and encoding latency across bit widths. 3 standard errors are shown.

**Colorlisting improves multibit performance**. Through colorlisting, we can take advantage of the surplus vocabulary partitions. Fig. 3 Left demonstrates the gain in the load capacity by using $r$=4 colorlists as opposed to $r$=2 given a fixed $\gamma$. We also show the results for $\gamma = .5$ and $r$=2. Besides the 8-bit case, which already achieves high accuracy, the performance of $\gamma$=.25, $r$=4 is statistically significant at p=$1e^{-2}$ than the second runner-up. We further discuss the implications of varying $\gamma, r$ in Section 5.

Next, we increase the number of tokens (T) and bit width accordingly to verify the effectiveness of embedding longer messages at a fixed bits per token. This resembles the scenario where the users generate longer sequences such as news articles or essays. While the message can be extracted up

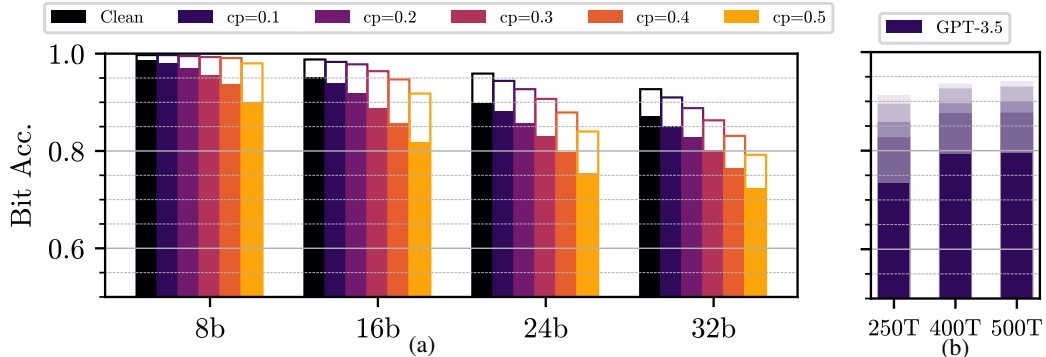

Figure 4: Corrupted bit accuracy for (a) copy-paste attack controlled by the human text percentage at T-250 and (b) paraphrasing attack using GPT-3.5 embedding 8-bit messages at varying token lengths. For (b), we show multiple sizes of list ($|L| \in \{2, 4, 8, 16\}$) by color gradation as 8-bit has relatively small output space.

to 90% accuracy up to 32-bit, the performance considerably falls for 64-bit. However, this can be partially compensated for by using list decoding as shown by the transparent bars, boosting the bit accuracy to more than 95% up to 32-bit.

**MPAC can maintain the watermark under corruption**. In the real world, a user may edit the generated text for better quality or in an attempt to evade the watermark. We study two types of attacks studied in the past work (Kirchenbauer et al., 2023b): *copy-paste* mixes the watermarked text and human text and *paraphrasing* uses another language model to paraphrase the watermarked text. Both attacks are realistic in that they do not maintain the start and end tokens of the watermarked text. The results in Fig. 4a demonstrate that for the copy-paste attack, the bit accuracy can be maintained to 90%(80%) for 8-bit (16-bit). Once again, list decoding is particularly effective, increasing the bit accuracy by 6.1% absolute for the corrupted case.

For paraphrasing, we use GPT-3.5.[4] We found paraphrasing to be much more challenging than the copy-paste attack and thus, experimented with only 8-bit messages and increasing the token lengths (Fig. 4b). With T=500, the bit accuracy reaches nearly 80% and with 16-list decoding, we are able to attain 90% bit accuracy across all token lengths. More attacks are considered in Appendix A.5.

**Detection performance is affected by bit-width.** To get a clearer picture of the detection performance, we compute AUC vs. the number of tokens observed in Fig. 5a following Kirchenbauer et al. (2023b). We see that the detection performance decreases as the message bit is increased. This phenomenon is similarly observed in other works as the increase in the number of "hypotheses" required to check leads to an increase in the false positive rate (Fernandez et al., 2023b). We further discuss the reasons behind this in the subsequent section. Note, however, that a watermarked text with 32-bit message reaches AUC over 0.99 once observing 200 tokens. The TPR at FPR=$1e^{-3}$ for b=$\{0, 8, 16, 24, 32\}$ are 0.98, 0.98, 0.95, 0.93, and 0.91, respectively (shown in Table 7).

**Across Model Scales, Datasets, Hash Schemes.** The results for larger models (13B, 70B) and other datasets are in Appendix A.6. To summarize, we found that text distributions with low entropy inherently have lower load capacity as observed similarly in Lee et al. (2023); Kirchenbauer et al. (2023b). We also present results for using another hash scheme with a longer context width in Appendix Table 10 and 11, which shows a similar level of performance.

**Comparison with Other Works.** We compare MPAC with Fernandez et al. (2023a, FCT) and Wang et al. (2023b, CTWL) in Table 1. For FCT, two zero-bit watermarking schemes are tested: Greenlisting Kirchenbauer et al. (2023a, Greenlist) and exponential minimum sampling Aaronson & Kirchner (2023, EMS). For FCT+Greenlist, using the Llama-2-7B tokenizer upperbounds the bit-width roughly to 15 bits. In summary, the clean performance and robust performance are similar in low bit-widths, but MPAC starts to outperform at 16-bit. FCT noticeably suffers from latency overhead when bit-width increases: increasing the bit-width from 16→24, lengthens the generation time by roughly 3.5x (14 seconds → 50 seconds) per sample. Decoding latency (message extraction)

---

[4]We use the most challenging prompt found for zero-bit watermarking (shown in Table 13)

| Copy-Paste ($p$) | B=8,T=250 | | | | B=16,T=250 | | | |
|---|---|---|---|---|---|---|---|---|
| | Clean | cp=10% | cp=30% | cp=50% | Clean | cp=10% | cp=30% | cp=50% |
| Ours | .986 (.06) | .981 (.07) | .956 (.10) | .900 (.13) | .951 (.07) | .939 (.08) | .887 (.09) | .819 (.12) |
| FCT+EMS | .979 (.10) | .943 (.17) | .858 (.24) | .800 (.28) | .905 (.20) | .811 (.26) | .702 (.26) | .601 (.23) |
| CTWL | .977 (.11) | .973 (.12) | .951(.16) | .858(.24) | .936 (.18) | .909 (.20) | .810 (.26) | .614 (.22) |
| FCT+Greenlist* | .995 (.05) | .988 (.08) | .970 (.12) | .908 (.20) | .986 (.09) | .974 (.12) | .929 (.18) | .765 (.26) |

Table 1: Comparison of multibit watermark performance with other methods on clean and corrupted settings. For corruption, we use the copy-paste attack. *The load capacity of FCT+Greenlist is limited to 15-bit.

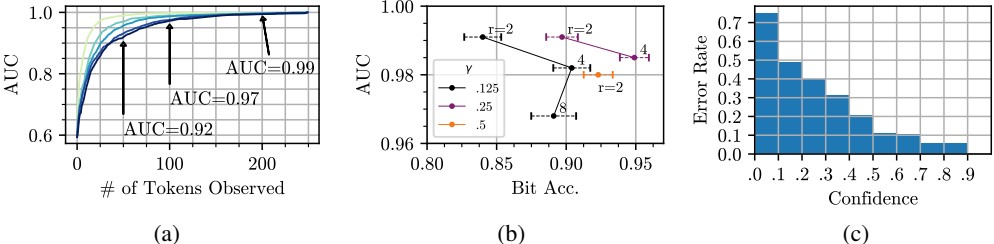

(a)            (b)            (c)

Figure 5: (a) AUC@number of tokens observed for $b=\{0, 8, 16, 24, 32\}$. Darker colors denote larger bit-widths. (b) Zero-bit and multi-bit watermark performance for varying $\gamma$ and $r$ for 1000 samples at T=100,b=8. (c) Error rate as a function of confidence.

is increased by 130x (0.87 second $\rightarrow$ 115 seconds) per sample. For more analysis, see Appendix A.4.

## 5   DISCUSSIONS

**Load capacity and detection performance trade-off.** As noted above, embedding longer messages degrades the watermark detection performance due to overestimating the statistics of non-watermarked human texts (Fig. 10). This is because computing the statistics involved finding the maximum cell value for each position. One natural solution is to use a better statistic that models the maximum cell value of a multinomial distribution. Empirically, we found that this performed on par or even slightly worse compared to the current approach, which may be due to the approximation error when using a small sample size. We give a more detailed discussion on this in Appendix A.8.

**Radix and Colorlist proportion** How do radix and colorlist proportion $\gamma$ influence multi-bit watermark performance? For $\gamma$=.125, the benefits of enlarging $r$ to 8 are saturated and show no statistical significance to $r$=4. While larger $r$ allows more tokens to be assigned to each position by reducing the effective length of the message, it challenges the problem by increasing the number of possible answers (digits) per position. Additionally, we observed that increasing radix trade-offs zero-bit performance for multi-bit performance. The observations are illustrated in Fig. 5b.

**List Decoding Ablation** In Fig. 5c, we show a plot of bit error rate stratified by confidence. While not properly calibrated (under-estimation), having higher confidence definitely shows the error rate is lower. We also highlight the effectiveness of this technique by comparing it with randomly outputting candidate messages from scratch in Table 4. We also observed that randomly altering a single position provides a good list as the best candidate message is already a good starting point.

## 6   CONCLUSION

Our findings demonstrate the viability of embedding any information into the outputs of language models while having the capability to distinguish between machine text and human text. This unveils a novel prospect of counteracting high-stake misuse of large language models via API. Furthermore, our analysis rooted in coding theory opens up other avenues for technical improvements such as using feedback or fusing error correction codes into MPAC. One limitation of our approach is the reduced separability of machine and human text when embedding longer messages. Overhauling this limitation can be a major step towards deploying multi-bit watermark in the real world.

## 7 ETHICS STATEMENT

Watermarking is one of the technologies that can mitigate malicious use cases by being able to trace back to the malicious user. However, ordinary users may find the idea discomforting as it may give the sense that the API provider can know what outputs are fed to the individual users. This is not the case unless the content is published to the public by the user, which – in many cases – is already done in an environment where the user can be identified (e.g. social media). All in all, the identification of machine-generated texts and tracing their provenance can enhance the accountability of API access of large language models without breaching individual users' privacy.

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

## A APPENDIX

**Table of Contents**

### A.1 DECODING ALGORITHM

---
**Algorithm 2:** Message Decoding

---
**Input:** Watermarked text $X_{1:T}$, hash context width $h$, effective message length $\tilde{b}$
**Output:** Predicted message $\hat{\mathbf{m}}$, number of colorlisted tokens $w$

    /* Initialize counter                                                */
1   $\mathbf{W}_p[m] = 0 \,\forall p, m$
    /* Count tokens in colored lists                              */
2   **for** $t$ *in* $[h+1, T]$ **do**
3      $s = f(X_{t-h:t-1})$
4      $p = \mathtt{sample}([\tilde{b}])$
5      **for** $m$ *in* $[r]$ **do**
6          Permute $\mathcal{V}_t$ using $s$ as seed
7          **if** $X_t \in \mathcal{G}_t^m$ **then**
8              $\mathbf{W}_p[m]$ += 1

    /* Predict message                                                    */
9   $\hat{\mathbf{m}}_r = $ " "
10   $w = 0$
11   **for** $p$ *in* $[\tilde{b}]$ **do**
12      $w$ += $\max(\mathbf{W}_p[m])$
13      $\hat{m} = \mathrm{argmax}_m(\mathbf{W}_p[m])$
14      $\hat{\mathbf{m}}_r$ += $\mathtt{str}(\hat{m})$
15   Get bit message $\hat{\mathbf{m}}$ by converting $\hat{\mathbf{m}}_r$
16   **return** $\hat{\mathbf{m}}, w$

---

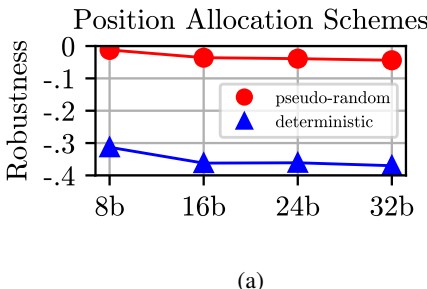 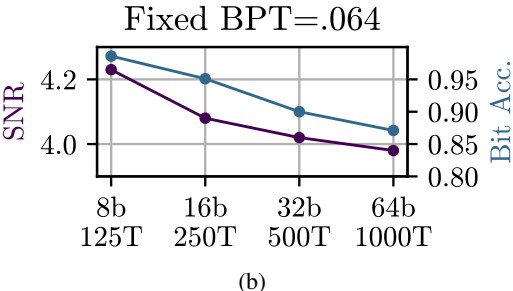

(a)               (b)

Figure 6: (a) Comparing robustness of the watermark (Clean performance - corrupted performance) for deterministic (cyclic) and pseudo-random position sampling schemes at T=250. (b) Relationship between bit accuracy and SNR for a fixed BPT.

## A.2 IMPLEMENTATION, HARDWARE, CODE DETAILS

We follow Kirchenbauer et al. (2023a) in most experimental settings. For the hashing scheme in the main paper, we use LeftHash scheme with context window $h = 1$. In the appendix, we provide results for the SelfHash scheme. For further discussions regarding the hash scheme see Appendix A.3. To generate sequences with the desired token length $T$, we generate with the max token set as $T$. Then we filter out the watermarked and non-watermarked sequences with token lengths under $T_{\text{low}} = T - \tau$. We set $\tau$=25, except for the LFQA dataset, which was set to $\tau$=50 as it has instructions that state to generate answers with 200-300 words. For generation, we use sampling with a temperature of 0.7. For each bit-width, a new set of generations had to be made as the length of the message differed.

For the copy-paste attack, we sample a random non-watermarked text and truncate to have the same length. Then, a position is randomly sampled to insert a $p$ percentage of the watermarked text into the non-watermarked text. We experiment with varying degrees of $p$ (10%∼ 50%).

We used `float16` for all our models during generation. Our experiment was run on a single NVIDIA A100. For T=250, generating around 500 watermarked and non-watermarked samples took approximately 200 minutes for the left hash scheme. When using the self-hash scheme, this took significantly longer (∼ 550 minutes). Our implementation is based on the official codebase of Kirchenbauer et al. (2023a): `https://github.com/jwkirchenbauer/lm-watermarking`. We will be releasing our code to reproduce our experiments.

## A.3 DISCUSSION ON THE HASHING SCHEME

The hashing scheme for generating the seed plays a significant role in watermarking. For our MPAC, the hashing scheme is employed once for position allocation and once for permuting the vocabulary list. Here, we discuss some implications of the design choices.

To recap, the function $f(X_{t-h:t-1})$ is used to hash $h$ most recent tokens before generating the $t^{\text{th}}$ token. Following the terminology of Kirchenbauer et al. (2023b), LeftHash takes the leftmost token, while SelfHash is determined in a slightly more complex way that is dependent on the $t^{\text{th}}$ token (see Algorithm 1 of Kirchenbauer et al. (2023b)). The context width and the hashing scheme determine robustness and quality (diversity) trade-offs. For our experiments, we use the two configurations (LeftHash with $h$=1 and SelfHash with $h$=4) proposed in the previous work found to be effective in the two aspects without further fine-tuning.

As expected by the trade-off, the perplexity was slightly higher for LeftHash compared to SelfHash (5.1 vs. 4.9 on average for 250 tokens), while P-SP was at the same level. One clear distinction between the two schemes was the encoding time latency. As SelfHash iteratively searches for tokens, this took significantly longer than the LeftHash scheme, which had nearly no overhead compared to no watermarking (appendix A.2 and Table 4). In addition, we observed that the sampled positions were not uniform for LeftHash with $h = 1$ as shown in Tab. 2 due to the reduced diversity of the tokens in the context width. Despite this, the multi-bit performance was similar for the two schemes

| | Ratio Sampled Position (Sorted) | | | |
|---|---|---|---|---|
| LeftHash ($h$=1) | 0.319 | 0.251 | 0.235 | 0.195 |
| SelfHash ($h$=4) | 0.264 | 0.257 | 0.242 | 0.238 |

Table 2: Ratio of the sampled position for $b$=8,$r$=4 (four positions total) for the two hashing schemes for position allocation.

(Table 10 and 11). A possible direction for improvement may be using different hashing schemes for position allocation (more robust) and vocabulary partitioning (more quality-focused).

### A.4 MORE RESULTS: LIST DECODING, LATENCY, FEEDBACK

**Comparison with Concurrent Works** In our experiments, we compared MPAC with Fernandez et al. (2023a, FCT) and Wang et al. (2023b, CTWL). Here we provide more details about the methodological differences and the experimental settings. We expect both methods to be relatively robust under realistic attacks that remove or add entire sentences as they rely on a pseudo-random function of the previous token(s) similar to ours. FCT uses message-specific secret keys to embed the watermark and is extendable to both Kirchenbauer et al. (2023a) and Aaronson & Kirchner (2023). For the zero-bit framework of Kirchenbauer et al. (2023a), they cyclically shift the greenlist $m$ times. This provides only $m$ distinct signals unique for each message. Since the size of the greenlist is equal to vocabulary size, the bit-width is bounded by. For the zero-bit framework of Aaronson & Kirchner (2023), the zero-bit watermark is embedded by using exponential minimum sampling, which relies on a random secret key (See A.10). In FCT, the random secret key of arbitrary size is created and cyclically shifted $m$ times. This allows embedding a larger bit-width than the aforementioned methods. As mentioned in the paper, both works have a computation cost that exponentially scales with the bit-width. Out of decoding and encoding, the encoding latency directly affects the API user, which is a crucial aspect for the API provider.

We generate 250 tokens on the C4 news-like subset, and the copy-paste percentage denotes the percentage of non-watermarked texts (higher denotes a stronger attack). We report the mean bit accuracy of roughly 500 samples. The results are in Table 1. Our results showed that all three methods achieved high performance in the 8-bit setting with FCT+Greenlist consistently outperforming all. However, as the bit-width and corruption rate increase, CTWL and FCT show a greater degree of degradation. For 16-bit, both (Fernandez et al., 2023a)+EMS and CTWL show considerable dropoff compared to ours. This shows that MPAC achieves superior robustness by separating the signals of each position. In contrast, the other methods display an all-or-nothing behavior where a message is either correct or completely random. While this may be beneficial in the high-performance regime (low bit-width and no corruption), this may lead to severe performance degradation when embedding longer messages. The distinction is also noticeable by the larger standard deviation of the two methods compared to ours.

**List decoding and Latency** We show absolute accuracy gained using confidence-based list decoding ($|L|$=16) compared with random decoding. We further compare the encoding and decoding latency for sequences with $\sim$ 250 tokens using a single Nvidia A100 when using an additive left hash scheme with context width 1. The results are in Table 4. The latency *does not* proportionally increase with message bit length, making it scalable to long messages. When using an efficient hashing scheme watermarking has a negligible increase in both encoding and decoding compared to vanilla generation, which requires 7.9 seconds and 0.09 seconds, respectively.

**Message Correction with Feedback** Here we provide some preliminary results of taking advantage of feedback during message encoding. One simple scheme is adapting the magnitude of the bias so that when the message is not correctly encoded, we enlarge the bias. Concretely, for $0 \leq t \leq T$ that is allocated to position $p$, if the current max colorlist does not match the actual message content, i.e. $\mathbf{m}[p] \neq \text{argmax}_j \mathbf{W}[j]$, we use a larger bias $\tilde{\delta} > \delta$. The results in Fig. 3 show that all lead to an increase in the multi-bit accuracy. However, we observed this came with a degradation in text quality measured by automatic metrics. We leave finding better methodology as a future work.

### A.5 MORE ON ROBUSTNESS: OTHER ATTACKS, DETECTION

| Accuracy Gained | | | | |
|---|---|---|---|---|
| | 8b | 16b | 24b | 32b |
| Confidence-based list | 1.1% | 3.7% | 6.0% | 5.6% |
| Random list | 0.6% | 0.4% | 0.5% | 0.3% |
| Latency (seconds/250 tokens ) | | | | |
| | 0b | 8b | 16b | 24b | 32b |
| Encoding (7.9) | 8.19 | 7.98 | 8.01 | 7.96 | 8.24 |
| Decoding (.09) | .08 | .09 | .09 | .09 | .10 |

Table 4: Comparison of absolute improvement in bit accuracy when using confidence-based list decoding and random list.

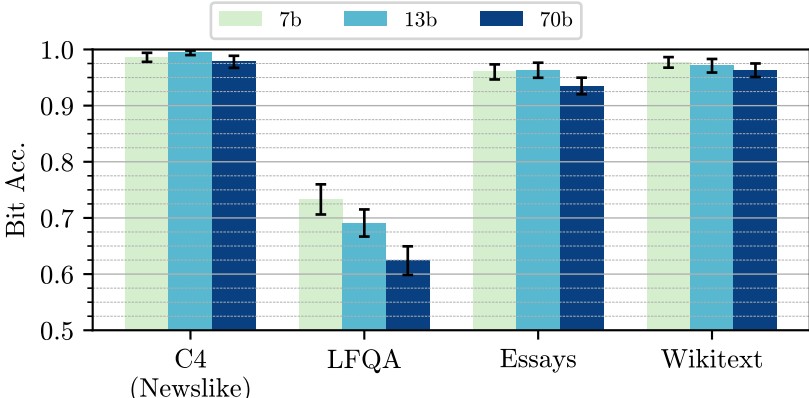

Figure 7: Multi-bit performance across datasets and model sizes.

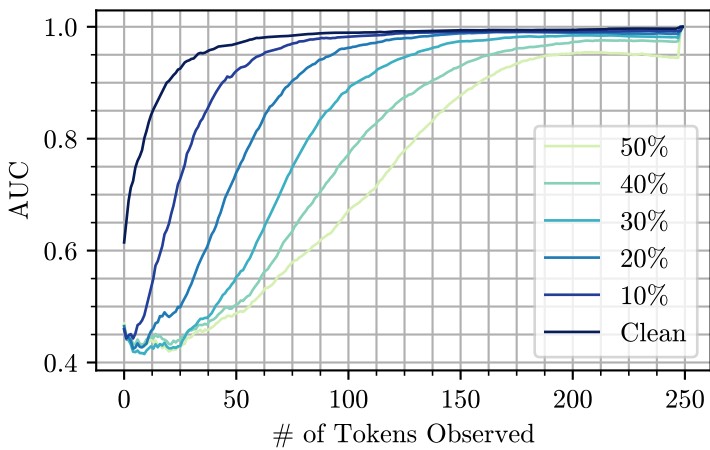

Figure 8: AUC vs. number of tokens observed when corrupted with copy-paste attack for 8-bit message.

| Bit Acc. after Paraphrasing with DIPPER | | | | |
|---|---|---|---|---|
| Bit-width | 8 | 16 | 24 | 32 |
| Best Prediction | .922 (.13) | .825 (.12) | .778 (.12) | .736 (.10) |
| 16-List Decoded | .982 (.05) | .924 (.08) | .864 (.10) | .801 (.09) |

Table 5: Robustness under paraphrasing using DIPPER (Lexical diveristy=20)

| | GPT-3.5 | DIPPER | | | |
|---|---|---|---|---|---|
| | | Lex.=20 | Lex.=40 | Lex.=60 | Lex.=60 Ordering=60 |
| P-SP | .815 | .933 | .897 | .844 | .827 |
| Absolute Change in # of Words | 36 | 13 | 16 | 19 | 20 |
| Bit Acc. | .733 | .922 | .849 | .757 | .719 |

Table 6: Comparison of the two paraphrasing method on text quality.

We also test our watermark against DIPPER (Krishna et al., 2023), which is a specialized paraphrasing model. DIPPER is parameterized by two scalers, which control lexical diversity and token order diversity. We first present the results across bit-width with a lexical diversity of 20 (out of 100). We see that the watermark fares considerably better than using GPT-3 attack in Table 5.

To see the magnitude of semantic drift of the two paraphrasing methods, we compute the P-SP between the original watermarked text and its paraphrased counter-

| Bit Accuracy | | | |
|---|---|---|---|
| $\delta$ | 0.5 | 1 | 2 |
| No feedback | .626 | .766 | .948 |
| $\tilde{\delta} = \delta + 1$ | .769 | .860 | .960 |

Table 3: Results for using feedback for adapting bias on T=100,b=8

part. We also compute the absolute change in the number of words. Table 6 demonstrates that paraphrasing using GPT-3.5 changes the semantic and the number of words greater than the setting used in Table 5, which may explain why the multi-bit watermark performance is lower for GPT-3.5. When we control the diversity parameters of DIPPER, this is able to degrade the watermark performance as well as GPT-3.5.

Some other forms of possible attacks considered in the literature are word substitution, insertion, and deletion. Word substition is very similar to the copy-paste attack considered in the main paper. Our watermark scheme is also robust to partial insertion and deletion of words as MPAC relies on the local context to synchronize the positions of the message and the ordering of the vocabulary.

**Robustness of zero-bit Watermark** Here we provide results for the detection performance under corrptuion. We use the copy-paste attack with the attack percentage ranges of {10%, 20%, 30%, 40%, 50%} and compare the AUC vs. number of tokens observed curve similar to Fig. 5a. While the detectability is noticeably affected, the final AUC is recovered to a large degree only after observing 250 tokens. In order of the attack strength, the final AUC's are .992, .987, .980, .971, .942, respectively. For the zero-bit counterpart, all the scores are over .990.

## A.6 ABLATIONS ON DATASETS AND MODEL SIZES

We show additional results on other datasets and model sizes in Fig. 7. C4 news-like subset is the dataset we used for our main experiment. "Long-form Question-Answering" (LFQA) is a dataset curated by Krishna et al. (2023) on the Reddit's "Explain Like I'm Five" (ELI5) forum. The Essays dataset comprises paris of instructions and essays (Schuhmann, 2022). Wikitext (Merity et al., 2016) comprises Wikipedia article. We use the 'wikitext-2' subset. For LFQA, we use the finetuned version, LLaMA-2-Chat, specialized for chats as they explicitly have questions or instructions as prompts.

It is apparent that the watermark performance is affected by the text distribution. When the entropy of the vocabulary distribution is low (low diversity), there is little room for encoding the message with a fixed bias, which has been observed in zero-bit watermarking as well where the watermark

performance suffers for low entropy text distributions such as coding (Lee et al., 2023; Kirchenbauer et al., 2023b). For our multi-bit case, this means the load capacity is inherently low for such text distributions. This is especially observed for LFQA, in which the model consistently starts the response by restating the question (e.g. *"The reason for [Question] is ..."*). Across the model scale, the trend is not as apparent although we found that the largest model consistently has a lower performance. This hints that the entropy of the vocabulary distribution is lower for the largest model, which might explain the higher text quality in general when we increase the model size. Larger models might have the capacity to form high-quality sequences even when the text distribution is altered by increasing the entropy via temperature or explicitly increasing the magnitude of the bias during watermarking. We leave this as a future work.

### A.7 METRICS: BIT ACCURACY, TEXT QUALITY

**Text Quality Metrics** Using P-SP, we measure the semantic similarity between the human text and watermarked text given the same prompt. While human evaluation is considered to be the golden label, our main purpose is to show that our multi-bit watermarking does not degrade the quality compared to zero-bit watermarking. Moreover, the effect of watermarking on the text quality *compared to no watermarking* shows promising results in human evaluations when sufficiently large models are used for open-ended generation by Kirchenbauer et al. 2023b (Appendix A.2 and A.9). Additionally, Fernandez et al. (2023a) also demonstrate that watermarking does not lead to noticeable performance degradation even on benchmarks with non-ambiguous answers such as coding

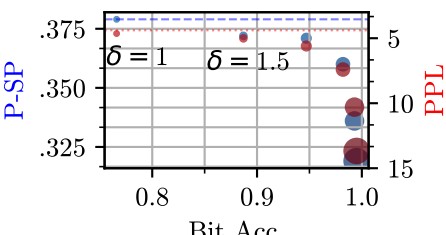

Figure 9: Text quality vs. $\delta$ across bias@T=100,b=8

and math especially with sufficiently larger models, albeit at a small bias. We further show in Fig. 9 the trade-off curve between bit accuracy and text quality. The size indicates the magnitude of bias ({1, 1.5 2, 3, 4, 5}) and horizontal dashed lines indicate non-watermarked counterparts. Analysis of text quality shows $\delta = 2$ lies at a good trade-off point.

**Bit Accuracy for Multi-bit Watermark** In our experiments, we used bit accuracy (error) as our metric for multi-bit watermark performance. This is a general metric that is independent of the downstream application or the encoding scheme. However, computing the exact match of a message should be done dependent on the context. To illustrate this, we start with some examples. First, consider the case where the encoding scheme to identify users is simply assigning a message to each user. Then, by embedding 4-bit message one can encode $2^4$ different users : **m**='0000' for Bob, **m**='0001' for Alice, and so on. For such a scenario, one might be interested in computing the exact match of the 4-bit message, also known as the packet error ratio. While this encoding scheme enables tracing back to the exact users at low load capacity, this is extremely inflexible as it cannot handle influx or outflux of users.

Conversely, one can turn to a more flexible encoding scheme by encoding each character. Using UTF-8, this requires 8 bits per character, which would mean 40 bits is required just for encoding 5 character user ID. For this scenario, one might be more interested in computing the packet error ratio of each character or the entire 40-bit message. A more realistic encoding scheme will be somewhere between the middle, which uses a more efficient representation, e.g. by merging often-used bytes as done in Byte pair encoding (Gage, 1994). Added with error correction codes such as the Reed-Solomon code (Wicker & Bhargava, 1999), this allows a more robust representation. Since focusing on a single type of encoding scheme – and more fundamentally, what information to embed – narrows down the potential applications, we present bit accuracy in our main experiments as done in previous works in the literature (Zhu et al., 2018; Luo et al., 2020; Yang et al., 2022; Yoo et al., 2023; Fernandez et al., 2023b). For T=250, the packet error ratio for the 8-bit message was 7.1%, which is +5.7 % higher than the bit error rate. With 16-list decoding, this is reduced to 2.4%.

Another metric considered in Table III of Fernandez et al. (2023a) was combining the detection scheme and packet error ratio. In this scenario, they assume using an encoding scheme of assigning each user to a single message (first example) and computing the percentage of finding the exact user

| | True Positive Rate | | | | |
|---|---|---|---|---|---|
| Bit-width | 0 | 8 | 16 | 24 | 32 |
| FPR=$1e^{-2}$ | 0.999 | 0.986 | 0.974 | 0.964 | 0.958 |
| FPR=$1e^{-3}$ | 0.997 | 0.974 | 0.956 | 0.943 | 0.915 |
| FPR=$1e^{-4}$ | 0.997 | 0.96 | 0.934 | 0.905 | 0.88 |
| FPR=$1e^{-5}$ | 0.994 | 0.951 | 0.907 | 0.851 | 0.793 |

Table 7: True positive rate at a fixed false positive rate across bit-widths. We use $\sim 500$ positive sample and $\sim$100,000 negative samples. We only count the unique tokens following (Kirchenbauer et al., 2023a; Fernandez et al., 2023a). This has an effect of removing outlier human text samples that have exceptionally high scores.

given a fixed false positive rate. At FPR=$1e^{-3}$ and using 8-bit message (256 users), we can correctly identify 92.6% cases. Our true positive rate was computed by the setting used in Table 7.

## A.8 ANALYSIS ON WATERMARK DETECTION

Here we further analyze how bit-width of the message and radix affect detection performance. Our analysis stems from the observation that as we increase the bit-width the detection score for the non-watermarked text increases more rapidly than that of the watermarked text. Consequently, the difference in the two scores *decreases* as larger bit-width is used, leading to reduced seperability. The results are in Fig. 10. Notice that the difference between the scores of watermarked and non-watermarked texts decreases for larger bit-width.

To grasp a hint of what is going on, we do away with the language model and other complexities by modeling this only through statistical distributions. To recap, our detection statistic (Eq. 1) was computed by aggregating the number of tokens in each position of the message. Letting $C_i$ as the number of tokens in the colorlist for the position $i$, we can write the aggregated form as

$$C = C_0 + \cdots + C_{p-1} \stackrel{H_0}{\sim} \text{Binomial}(T, \gamma) \tag{2}$$

However, note that during decoding the ground truth message is unknown and thus, is predicted by taking the colorlist that has the max number of tokens. This is problematic when decoding for non-watermarked text as it biases the statistic to be higher when bit-width is increased. We let $W_i = [w_0, \ldots, w_{r-1}]$ be the number of tokens in $r$ colorlists (strength of watermark) for position $i$. For a non-watermarked text, we can assume that this is a random variable with equal probability for each colorlist

$$W_i \sim \text{Multinomial}(n_i, [\gamma \cdots \gamma]) \tag{3}$$

where $n_i$ is the number of tokens allocated to position $i$. Our decoding method takes the maximum cell value of this, which makes itself a random variable:

$$W_i^{\max} = \max(W_i) = \max([w_0, \ldots, w_{r-1}]). \tag{4}$$

Our final statistic used for our detection score is the sum of this variable over the entire positions:

$$W^{\max} = \sum_i^p W_i^{\max} \tag{5}$$

We see that our statistic is dependent upon the number of candidates when selecting the maximum cell (i.e. radix) through Eq. 4 and the number of positions (i.e. bit-width) through Eq. 5.

To verify the effect of bit-width and radix on the detection score, we compare the difference in the statistics for a uniform multinomial distribution, which signify non-watermarked text, and a multinomial distribution with a slightly modified probability $[\gamma + \epsilon, \gamma, \ldots, \gamma]$ to signify the added bias term for the watermarked distribution. We sample 1000 samples of $W^{\max}$ and compute the difference in the detection scores for the two distributions. The results in Fig. 11 corroborate that an increase in bit-width / radix decreases the separability of the detection scores.

In an attempt to overhaul this, we tried computing the likelihood of $W_i^{\text{rm}}$ before aggregating them using an approximation of Levin (1981) (More details in the next section). However, this only led

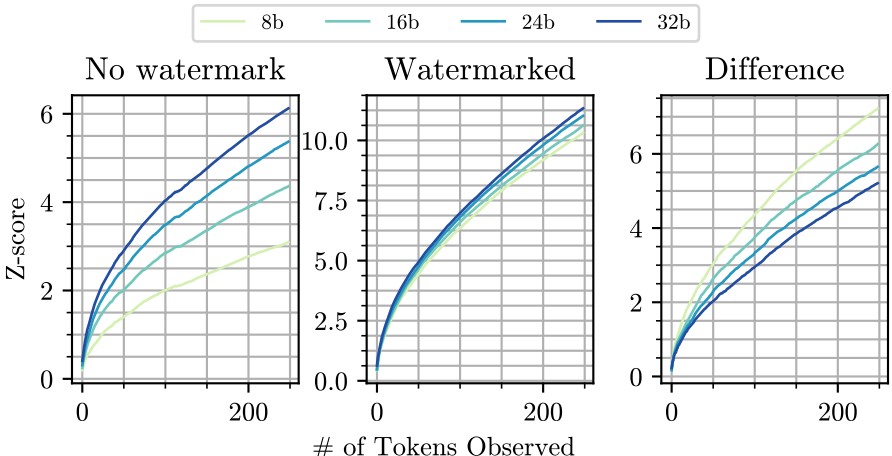

Figure 10: The detection scores of non-watermarked texts, watermarked texts and their difference as a function of number of tokens observed. We see that the difference in the scores decreases as bit-width increases, leading to reduced seperability.

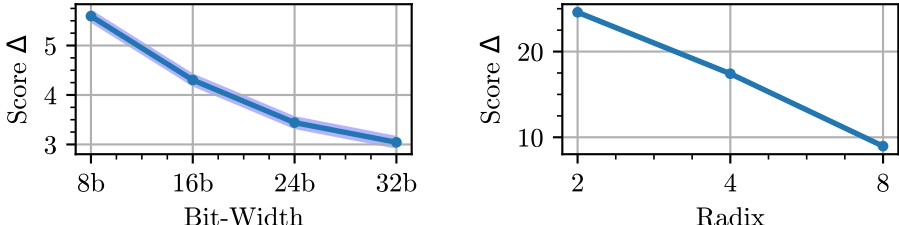

Figure 11: Simulation of the difference between (unormalized) scores for watermarked and non-watermarked multinomial distributions. Higher score signify higher seperability, hence higher detection performance. We use $\epsilon$=0.1. For right, we use $\gamma$=.125 to allow more radix.

to on par or slightly worse performance. This may be because $n_i$ is small for cases when $T$ is small compared to the length of the message. Other than this, some of the approaches we attempted were:

- Computing test statistic per position or weighting the statistic of each position with $n_i$ before aggregating.

- Computing the p-value of the binomial random variables rather than using the normal approximation, i.e. regularized incomplete beta function.

- Computing the p-value under the null hypothesis that the distribution of the colorlists follows a uniform distribution, i.e. Chi-square Goodness of Fit test

All the approaches either led to on-par or slightly worse results.

## A.9 APPROXIMATING MAX MULTINOMIAL CELL DISTRIBUTION

We used the approximation of Levin (1981) for modeling the distribution of the maximum cell frequency. For completeness, we present the steps used for the approximation adapted to our case. For a multinomial distribution with sample size $N$ and probability vectors $[p_0, \ldots, p_{r-1}]$, Let $a$ be the maximum cell value, then the cumulative distribution function of having a maximum value of $a$

can be approximated for any real number $s > 0$

$$P(a) = \frac{N!}{s^N e^{-s}} \{\prod_i^{r-1} P(X_i \leq a)\} P(W = N) \tag{6}$$

where $X_i \sim \text{Poisson}(sp_i)$ and $W = \sum_i^{r-1} = Y_i \sim \text{Truncated Poisson}(sp_i)$ with range $0, 1, \ldots, a$. Following Example 1 of Levin (1981), we set $s = N$ and use Stirling's approximation for $N!$. We also approximate $W$ using the normal approximation to the Poisson distribution.

## A.10 EXTENSION TO OTHER ZERO-BIT WATERMARKING

Aaronson & Kirchner (2023) is another line of work in zero bit watermarking that modifies the sampling process by generating a secret vector $\mathbf{r} \in [0, 1]^{|\mathcal{V}|}$ based on the random seed $s$. Given the original probability distribution $\mathbf{p}^{|\mathcal{V}|}$, the token with both large $p_v$ and $\mathbf{r}_v$ is favored by choosing

$$x = \text{argmax}_{v \in \mathcal{V}} \mathbf{r}_v^{1/\mathbf{p}_v}. \tag{7}$$

We can adapt our position allocation method to this as well by preceding the above step with position allocation. Then, the secret key can be modified depending on the message content by the following rule:

$$\mathbf{r} = \begin{cases} \mathbf{r} & \text{if } \mathbf{m}[p] = 0 \\ \mathbf{1} - \mathbf{r} & \text{if } \mathbf{m}[p] = 1 \end{cases} \tag{8}$$

where $\mathbf{1}$ is a vector with 1 in all the elements. Analogous to favoring mutually exclusive colorlists, this allows favoring different tokens depending on the message content. At decoding time, we can similarly maintain a counter for each position for the two cases.

## A.11 TABULAR RESULTS

Here we present the numerical results for the experiments done in the main paper. Numbers in the parenthesis signify the standard deviation.

- Table 8 $\leftrightarrow$ Figure 9 show the relationship between $\delta$ vs. text quality and watermark strength.
- Table 9 $\leftrightarrow$ Figure 3 left compare the different configurations of radix and colorlist proportion.
- Table 10 $\leftrightarrow$ Figure 3 left show the multibit watermark performance on a fixed token length.
- Table 11 $\leftrightarrow$ Figure 3 right show the multibit watermark performance on a fixed load capacity (bits per token).
- Table 12 $\leftrightarrow$ Figure 4a show the multibit watermark performance under copy-paste corruption.
- Table 13 $\leftrightarrow$ Figure 4b show the multibit watermark performance under paraphrasing.

| $\delta$ | 0.5 | 1 | 1.5 | 2 | 3 | 4 | 5 |
|---|---|---|---|---|---|---|---|
| Bit Acc. | .626 (.19) | .766 (.18) | .887 (.15) | .947 (.11) | .982 (.08) | .993 (.05 | .995 (.05) |
| P-SP (w/ reference) | .385 (.15) | .379 (.15) | .372 (.15) | .371 (.15 | .360 (.14) | .336 (.13) | .319 (.13) |
| P-SP (w/ non-wm.) | .526 (.18) | .460 (.16) | .433 (.15) | .417 (.15) | .388 (.14) | .349 (.14) | .330 (.13) |
| PPL | 4.41 (1.5) | 4.64 (1.8) | 5.01 (2.0) | 5.6 (2.0) | 7.41 (2.7) | 10.3 (4.1) | 13.67 (5.9) |

Table 8: Bit accuracy and text quality on embedding 8 bit-width message on T=250 across various magnitudes of bias $\delta$.

| Bit Accuracy @ T=250 | | | | |
|---|---|---|---|---|
| Bit | 8 | 16 | 24 | 32 |
| $\gamma=.25, r=4$ | .986 (.06) | .951 (.07) | .900 (.09) | .871 (0.08) |
| $\gamma=.25, r=2$ | .966 (.07) | .905 (.08) | .858 (.08) | 0.820 (.08) |
| $\gamma=.50, r=2$ | .978 (.05) | .922 (.07) | .875 (.08) | 0.849 (.07) |

Table 9: Multibit watermark performance measured by bit accuracy for varying configurations of colorlist proportion and radix.

| Bit Acc. @ T=250 | | | | |
|---|---|---|---|---|
| Bit | 8 | 16 | 24 | 32 |
| LeftHash($h=1$) | .986 (0.06) | .951 (.07) | .900 (.09) | .871 (0.08) |
| SelfHash($h=4$) | .976 (.08) | .905 (.08) | .895 (.09) | .862 (.09) |

Table 10: Bit accuracy for two different hash schemes for a fixed token length.

| Bit Acc. @ BPT=.064 | | | | | |
|---|---|---|---|---|---|
| T | 63 | 125 | 250 | 500 | 1000 |
| Bit | 4 | 8 | 16 | 32 | 64 |
| LeftHash($h=1$) | .961 (.13) | .958 (.09) | .951 (.07) | .913 (.08) | .846 (.09) |
| SelfHash($h=4$) | .952 (.13) | .953 (.10) | .945 (.08) | .911 (.08) | .850 (.08) |

Table 11: Bit accuracy for two different hash schemes for a fixed bits per token.

| Copy-paste Attack | | | | | | | |
|---|---|---|---|---|---|---|---|
| Attack Strength | | Clean | 10% | 20% | 30% | 40% | 50% |
| 8-bit | Best | .986 (.06) | .981 (.07) | 0.971 (.08) | .956 (.10) | .938 (.12) | .900 (.13) |
| | +16-List | .997 (.02) | .997 (.02) | .995 (.03) | .993 (.03) | .991 (.04) | .980 (.05) |
| 16-bit | Best | .951 (.07) | .939 (.08) | .918 (.09) | .887 (.09) | .858 (.11) | .819 (.12) |
| | +16-List | .988 (0.04) | .983 (.04) | .978 (.05) | .964 (.06) | .947 (.07) | .918 (.08) |
| 24-bit | Best | .899 (.09) | .882 (.09) | .858 (.10) | .830 (.10) | .797 (.11) | .755 (.11) |
| | +16-List | .959 (.06) | .944 (.06) | .927 (.08) | .907 (.08) | .879 (.09) | .840 (.09) |
| 32-bit | Best | .871 (.08) | .851 (.09) | .828 (.09) | .801 (.09) | .765 (.09) | .723 (.1) |
| | +16-List | .927 (.07) | .910 (.08) | .888 (.08) | .863 (.08) | .831 (.09) | .792 (.09) |

Table 12: Robustness when certain percentage of human text is mixed into the watermarked text.

| GPT-3.5 Paraphrasing | | | | |
|---|---|---|---|---|
| Token Length | | 250T | 400T | 500T |
| | Best | .733 (.19) | .792 (.19) | .795 (.19) |
| | +2-List | .825 (.16) | .874 (.15) | .875 (.15) |
| 8-bit | +4-List | .856 (.14) | .894 (.13) | .898 (.13) |
| | +8-List | .893 (.12) | .924 (.11) | .928 (.11) |
| | +16-List | .911 (.10) | .934 (.09) | .939 (.09) |

**Prompt**

"As an expert copy-editor, please rewrite the following text in your own voice while ensuring that the final output contains the same information as the original text and has roughly the same length. Please paraphrase all sentences and do not omit any crucial details. Additionally, please take care to provide any relevant information about public figures, organizations, or other entities mentioned in the text to avoid any potential misunderstandings or biases."

Table 13: Robustness when paraphrased with GPT-3.5.

A.12 GENERATION SAMPLES

| Prompt | Non-Watermarked | Watermarked |
|---|---|---|
| A blind man in his 60s is searching for the young man who pulled him back from an approaching train and saved his life last Thursday, Nov. 9. Mike Wyatt stood at the stairs to the Peoria train station in Aurora, Colorado, on Monday, looking for the young man who saved his life days earlier | "He told me he was 20 years old and he was going to college," Wyatt told reporters. "He said he was going to be a doctor." Wyatt, who is blind and has been for most of his life, was at the train station ready to catch a train to the airport when he slipped and almost fell in front of an oncoming train.[...] | and has given him new life. Inside the station, he spoke to the counter clerk, hoping the clerk might have information that might lead him to the young man, who jumped at the chance to help Wyatt when he saw he was in danger. But, he said, he found nothing.[...] |
| [...] Shortly after the release of the lead single "Life's About to Get Good," it hit number one on the Billboard 200. Of course, fans were simply dying to see how their favourite country icon was doing, so | the album was released on CD, vinyl, and digital. But, Shania took things to the next level and released a deluxe version with added tracks. [...] | the song was released a few months earlier in June. "OK so I am sooooo proud that my song Life's About to Get Good is now the number one song on Billboard country," she tweeted back in June.[...] |
| [..] Amazon's offices will be located within an already-established special tax district where a portion of the property tax revenue goes toward infrastructure improvements such as parks and wider sidewalks. The incentive agreement says that half of any new revenue from that district starting in 202 | 3 would go to Amazon to help pay for employee benefits and transportation, but the county has said it would not be required to pay those benefits.[...] | 7 would go to Amazon through a 15-year payment schedule. This amount would grow to more than 60 percent if the company exceeds 10.24 million square feet of office space. Amazon is expected to reach that level in 2027.[...] |
| [...] "Below the surface of the Sun a dynamo process is working creating magnetic field," Harra explained in an email interview. "When this becomes buoyant it can rise to the surface of the Sun, | and this is what we call a sunspot." The new research shows that the initial explosion was actually a series of smaller blasts that combined to create the X-7 flare. | and form coronal holes and sunspots. This happens in a fairly homogeneous area on the Sun, so there can be several sunspots with a single magnetic field underneath. |
| [...]The merge listing the most important changes to Linux 3.8's sound subsystem includes some other changes to audio drivers. The kernel now includes a driver for human interface devices (HIDs) that use I2C (1, 2 and others), using the "HID over I2C" protocol designed by Microsoft and implemented in WindowsÂ | 7 and later versions of the operating system. The kernel now has a driver for the Samsung Galaxy S III smartphone's touchscreen (1, 2 and others), and the rt2800usb driver, for the RaLink RT2800USB WLAN chip, now supports devices that have Bluetooth 3.0 (1, 2).[...] | 7 and Windows Vista. The drivers can read out data from HIDs and set the appropriate commands to them. An example of such a device is a BT-USB adapter. The sound subsystem now supports two new, high-quality audio codecs (1, 2):[...] |

Table 14: Randomly sampled examples of watermarked texts on the C4 newslike subset with 100% bit accuracy. Samples are truncated for readability.

