# OpenReview forum: "Advancing Beyond Identification: Multi-bit Watermark for Large Language Models"
_ICLR.cc/2024/Conference — ICLR 2024 Conference Withdrawn Submission_

### Official Review · Reviewer_Fyz9 · 2023-10-22

**Soundness:** 2 fair
**Presentation:** 2 fair
**Contribution:** 2 fair
**Rating:** 3
**Confidence:** 4

**Summary:**

In this paper, the authors proposed a new multi-bit watermark for language models. This strategy ensures the robust extraction of extensive watermarks without requiring model access or fine-tuning, preserves text quality, supports zero-bit detection, and withstands intensive adversarial alterations such as text interleaving and paraphrasing.

**Strengths:**

The introduction of a multi-bit watermark presents an innovative concept within this research domain.

Experimental results affirm the robustness of the proposed watermark algorithm.

**Weaknesses:**

1. The multi-bit watermark methodology, at a cursory glance, seems akin to an expansion of the red-green list from [1] into a more nuanced multi-color list. Given that the detection methodologies remain consistent, the paper should elucidate how the multi-bit approach enhances the robustness intrinsic to [1]. As this constitutes the crux of the paper's contribution, it is imperative for the authors to furnish theoretical insights and a robust analysis concerning the multi-bit watermark's robustness.

2. There is a noticeable absence of a comparative robustness analysis between the proposed method and that established in [1] within the experimental evaluations. Considering the resilience of the [1] watermark against text alterations, it is essential for the authors to demonstrate superior robustness to substantiate their contributions effectively.

3. While the authors purport that the multi-bit watermark sustains the caliber of the text, the empirical evidence, particularly in Table 1, suggests parity in quality between texts generated by the multi-bit and zero-bit watermarks. It is incumbent upon the authors to extend their comparison to texts generated without watermarks to convincingly affirm that the multi-bit watermark preserves text quality.

4. Figure 5(a) depicts a correlation wherein an augmentation in bit-width correlates with diminished robustness. This trend ostensibly advocates for the utilization of the zero-bit watermark, prompting the question of whether, in terms of robustness, [1] might present a more formidable solution than the proposed multi-bit watermark.

[1] A Watermark for Large Language Models. John Kirchenbauer, Jonas Geiping, Yuxin Wen, Jonathan Katz, Ian Miers, Tom Goldstein. ICML (2023)

**Questions:**

See Weaknesses

---

> ### Author Response · Authors · 2023-11-15
>
> Thank you for taking your time to review our paper. We hope our response will clarify any misunderstanding of where our paper stands. If you have further questions that prevent you from raising your score, please let us know. For readability, we will put the contents identical to the general response in block quotes.
>
>
> **the paper should elucidate how the multi-bit approach enhances the robustness intrinsic to Kirchenbauer et al., (2023a)**
> >This was addressed in our general response, but we will repeat the points. We respectfully disagree with this point. Our work does not claim to “enhance the robustness intrinsic to” Kirchenbauer et al., (2023a). Rather, we propose a new research direction for extending this to multi-bit. Comparing the performance directly with KGW is not an apples-to-apples comparison. In Section 3.2 Message Encoding, we showed that the zero-bit detection problem is equivalent to embedding a single-bit. As watermarking (Vleeschouwer et al., 2002) involves a tradeoff between load capacity (how much information) and robustness (how much information is preserved), we tackle an inherently more challenging task. Nonetheless, we believe providing a comparison does make the work more complete. As such, we discuss in detail the tradeoff between message length and detection in Section 5 and present FPR comparison with zero-bit detection in Table 6.
>
> Perhaps the reviewer could be asking to compare the watermark detection performance against Kirchenbauer et al., (2023a) to quantify the degradation as bit-width increases. This is expected as the zero-bit setting is a special case of embedding a single-bit message of $\mathbf{m}=$”0” in all the samples. To give an apples-to-apples comparison, we embed the __same__ 8-bit message ($\mathbf{m}$=”00000000”). This is essentially the same setting with the zero-bit watermarking, but applying only Position Allocation on top of it. To see the true positive rate at lower FPR, we sampled 10,000 negative samples and computed the true positive rate. Below, we show the results for zero-bit and 8-bit.
> |Bit-width|@FPR=1e-3 | @FPR=1e-4| @FPR=1e-5|
> |---|--|--|--|
> |0| 0.997|0.997|0.997|
> |8 (single message)| 0.994|0.992|0.992|
> |8 |0.974|0.965|0.965|
>
> As expected, embedding the same message using 8-bit shows nearly identical results. Compared to this, actually embedding the unique 8-bit message does lead to performance degradation, but still maintains a fairly high TPR even at FPR=1e-5.
>
> In Appendix A.8, we discuss in depth the reason behind this is that the detection metric of non-watermarked (human) text is over-estimated as bit-width is increased. In the revised paper, we will mention this limitation. We leave improving the detection methodology for future research.

---

> ### Author Response · Authors · 2023-11-15
>
> **noticeable absence of a comparative robustness analysis between the proposed method and that established in Kirchenbauer et al., (2023a)**
>
> We disagree. While the mentioned work discusses a variety of possible attacks, which is in itself valuable, the only empirical result is Figure 6 which experiments with paraphrasing using T5. We follow the experimental setting of Kirchenbauer et al., (2023b), which focuses on the robustness of the zero-bit watermark. We show the results on three realistic and strong attacks:
>
> -	Copy-pasting human text (Figure 5a)
> -	Paraphrasing using Chat-GPT (Figure 5b)
> -	Paraphrasing using a specialized model (Table 4)
>
> Reviewer `hfyX` commented that our paper contains thorough experiments with multiple sets of attacks, and Reviwer `eLnz` commented that “one of the strengths of the paper is its demonstration of the model's performance under copy-paste attacks”.
>
>
> We believe our paper would be strengthened with an additional comparison with other _multi-bit_ methods. As far as we know, the most robust multi-bit watermarking method among post-processing methods (Abdelnabi & Fritz, 2021, Yang et al., 2022)  is Yoo et al. (2023). However, this work only considers corruption at word-levels such as deletion, insertion, and substitution. Methodologically, one clear limitation is the sentence-level method, whereby embedding and extraction are done at each sentence level. As an illustration, consider the example below:
> ```
> Sentence 1 -> extracted message : “000”
> Sentence 2 -> extracted message : “000”
> Sentence 3 -> extracted message : “111”
> Sentence 4 -> extracted message : “111”
> Final extracted message : “000000111111”
> ```
> If any human text were to be added into the first part of the writing (instance of copy-paste attack), the message is irrecoverable as it has no way to identify _when_ the machine text starts or ends. The problem persists when a single sentence in the middle is removed as well. In contrast, the proposed algorithm MPAC need not know this and achieves relative robustness in various kinds of attacks.
> >The concurrent works released around July 2023, which are listed above in the general comments and Related Works section of the paper, also rely on the hashing function similar to ours. We expect them to have better robustness than the post-processing methods. We are working on presenting the comparison of the robustness and load capacity of these methods in the following days. Theoretically, our MPAC has an advantage in embedding long messages without any added latency during generation (Figure 3).
>
> **While the authors purport that the multi-bit watermark sustains the caliber of the text, the empirical evidence, particularly in Table 1, suggests parity in quality between texts generated by the multi-bit and zero-bit watermarks. It is incumbent upon the authors to extend their comparison to texts generated without watermarks to convincingly affirm that the multi-bit watermark preserves text quality.**
>
> >As stated in the general response, we do not “_purport that the multi-bit watermark sustains the caliber of the text_” compared to non-watermarked text. As stated in our introduction and Figure 3, we state that our text quality is maintained compared to zero-bit watermarked texts. We will revise the wording to avoid any further confusion.
>
> We are unsure of which table you are referring to as Table 1 is unrelated to text quality. If you have any reason to believe otherwise, please let us know. We will try to clarify this as soon as possible.
>
> **Figure 5(a) depicts a correlation wherein an augmentation in bit-width correlates with diminished robustness. This trend ostensibly advocates for the utilization of the zero-bit watermark, prompting the question of whether, in terms of robustness, [1] might present a more formidable solution than the proposed multi-bit watermark.**
>
>
> Yes. We showed that the increasing bit-width does affect the performance. However, we disagree that this leads to the conclusion that Kirchenbauer et al., (2023a) “might present a more formidable solution than the proposed multi-bit watermark.”. Embedding multi-bit information opens up a wide array of applications previously infeasible by zero-bit watermark alone. Moreover, we discuss and analyze in depth the limitations of multi-bit watermarking in (1) Section 3.3 and Figure 2b how SNR is a more reliable factor in estimating the performance and (2) the tradeoff between the capacity of the watermark and its detection rate (Section 5 and Appendix A.8.  We believe this discussion is an important contribution in enhancing the performance in the future.
>
>
> ## Reference
>
> Kirchenbauer, John, et al. "A watermark for large language models." arXiv preprint arXiv:2301.10226 (2023).
>
> Kirchenbauer, John, et al. "On the Reliability of Watermarks for Large Language Models." arXiv preprint arXiv:2306.04634 (2023).

---

### Official Review · Reviewer_hfyX · 2023-10-23

**Soundness:** 3 good
**Presentation:** 4 excellent
**Contribution:** 2 fair
**Rating:** 6
**Confidence:** 5

**Summary:**

This paper proposes a scheme for “multi-bit watermarking” texts generated by language models by extending a work on “zero-bit” watermarking by Kirchenbauer et al. [2023a].

---

## Comment After Rebuttal

Thank you for taking the time to address all my concerns and answering all my questions!

I would appreciate it if the author(s) could incorporate in the revised version my suggestion regarding the steganography, TPR/FPR of the watermark "detection," and other clarifying remarks the author(s) have provided. After reading the other reviews, I also echo the importance of reporting the results on instruction-tuned models. I still would like to advocate for the slightly different formulation of the "list decoding" I proposed, but that's not a big deal.

Overall, the main weaknesses of the paper are the lack of technical novelty and the less-than-ideal empirical results. Regardless, I believe that my original review may have been too harsh on these aspects. The author(s) have convinced me to see the practical value of the problem and the fact that they are one of the first to study it. Given these facts, I believe that the weakness in terms of the empirical results is not so significant, and this line of work is worth iterating on in the future. As a result, I'm willing to raise my rating from 5 to 6.

**Strengths:**

### Originality

The proposed method is relatively simple and is a nice extension of the zero-bit watermarking scheme. This means that any intuition as well as tools and analysis that are based on Kirchenbauer et al. [2023a] should also carry over and benefit the understanding of this scheme.

### Quality

This paper contains thorough experiments considering multiple sets of parameters, models, metrics, and attacks. There is a good amount of numerical results and textual analyses in both the main text and the appendix, making the results convincing.

### Clarity

The paper is well-written and easy to follow. Section 3 and Figure 1 provide a simple description of the scheme. The messages are effectively conveyed; all the figures and tables are easy to interpret.

**Weaknesses:**

### Significance

Watermarking for detecting machine-generated content is an intellectually interesting work, but I’m not sure about its practical benefits for two reasons. First, with open-source LLMs, it is virtually impossible to enforce the “post-processing” watermarks. Second, under the assumption that there’s an adversary trying to remove the watermark (e.g., via back-translation or paraphrase), it is very difficult to achieve a satisfactory TPR with a low enough FPR. This problem is heavily asymmetric in the sense that the cost of an FP is gigantic (wrongly detecting a human-written text as a machine’s) compared to an FN so an acceptable FPR has to be very low in practice (~1e-5).

### Comparison to Steganography

The paper briefly mentions steganography and claims that it is different from watermarking. However, I’m not convinced that they are different. The undetectability property of steganography is exactly desired by LLM watermarking, i.e., a watermark should not modify the distribution of the “cover text” (or LLM-generated text) in a noticeable way. Generally, steganography and watermarking differ in their purpose, but in the instance of LLM, they are identical concepts. So it seems important that this paper (and other watermarking papers) is **compared (theoretically and/or empirically) to steganography on natural language** such as [1] and [2].

### Methodology

- List decoding can be better parameterized by a confidence threshold rather than a fixed length $|\mathbb{L}|$ like a 95%-confidence interval. More precisely, given a confidence threshold (says 95%), we want to make a statement like: “With probability 95%, the true message is among these five messages” (think of [conformal prediction](https://en.wikipedia.org/wiki/Conformal_prediction), for example). There should be a principled way to either model this directly or compute this by aggregating the confidence score at each position ($c_i$)—This can be a bit tricky since $c_i$’s are not independent.
- Also including the list decoding as part of the bit accuracy metric (e.g., in Figure 4 and 5) seems irrelevant and unjustified to me. It’s not clear how the practitioners would be able to utilize the 16 other less-confident messages in the LLM detection/attribution setting.
- One very nice property of the zero-bit watermarking scheme proposed by Kirchenbauer et al. [2023a] is the theoretical guarantee on the number of tokens in the green list (Theorem 4.2). It would be nice to see a similar theoretical analysis on this paper.

### Experiments

**Metric.** AUC is a relatively misleading metric in practice for watermarking. As mentioned earlier, each FP is very costly so an appropriate metric in this scenario is usually something like TPR at a very low level of FPR (e.g., 1e-2, 1e-4, 1e-6, etc.).

**Comparison to zero-bit watermarks.** I would like to see a comparison of the efficiency of this multi-bit watermark to the previously proposed zero-bit ones (at least, Kirchenbauer et al. [2023a]) for just detecting machine-generated texts. This is important because prior to attributing the text to which model/service, we have to first decide whether the text is generated by a machine or a human. In theory, this scheme should trade off the ability to encode longer messages with the watermark detection efficiency.

### Other Minor Issues

- Figure 2 caption: “Clean bit error - corrupted bit error” was a little bit unclear as the metrics have not been defined at this point.
- [Typo] Page 9: “**Across Model Scales, Datasets, Hash Schemes.** The results for larger models (13B, 70B) and other datasets are in Appendix A.4.” should be Appendix A.6?

References
- [1] https://arxiv.org/abs/1909.01496
- [2] https://arxiv.org/abs/2210.14889

**Questions:**

- Page 4: How exactly is the hash function $f$ applied to the previous tokens (step 1 on page 4)? What’s the hash length $h$? How does $h$ affect the robustness or the bit accuracy?
- Page 6: In “**List Decoding** The linear nature with respect to…”, what does “linear nature” refer to?
- Page 7: In the last paragraph, it is unclear how the scheme works when $\gamma \ne 1/r$ (e.g., $\gamma = 0.25$ and $r=2$).
- Page 7: In the last paragraph, how is the $p$-value computed? Why is it stated relative to the runner-up instead of a null hypothesis? I’m not sure if this has been mentioned previously.
- Figure 4: Why does $r=4,\gamma=0.25$ perform better than $r=2,\gamma=0.5$ when $b$ and $T$ are fixed? Detecting two “options” should be easier than four. Is this due to the effective bit-width (i.e., the message when $r=2$ is actually twice as long as when $r=4$)?
- Page 8: Where can I find more detail on how the copy-paste attack was carried out?

---

> ### Author Response · Authors · 2023-11-14
> **Responses to Weaknesses [1]**
>
> Thank you for the very detailed feedback. We tried our best to address your concerns below. Please let us know if there are remaining concerns that prevent you from raising your scores.
>
> ## Significance
>
> Please note the next paragraph is nearly identical to the general response.
>
> The first point raised by the reviewer was that the current decoding-based watermarks cannot be enforced to open-sourced language models. As answered in the general response, we, nonetheless, believe studying watermarking methods for close-sourced language models is important because of its prevalence. The largest API provider, ChatGPT, had over 206 million monthly unique visitors in April, while Bard had over 140 million monthly unique visitors in May. Moreover, companies in Korea and China have also launched servicing their close-sourced models as API or apps (Baek, 2023; David, 2023). As of now, the most capable models are arguably close-sourced and if this trend were to continue, it is likely that the number of users will not decline. Watermarking is one of the solutions that can mitigate malicious uses of closed-sourced language models. Having a reference point in academia strongly incentivizes the owners of closed-sourced models to apply them.
>
> The second point raised by the reviewer was regarding achieving a satisfactory TPR with a low enough FPR. We agree. However, we would like to note that our work is one of the first works directly watermarking language model outputs along with the two concurrent works uploaded in July 2023 (Fernandez et al., 2023a; Wang et al., 2023). In fact, watermarking language model outputs in a zero-bit manner (Kirchenbauer et al., 2023) was proposed less than a year ago. We believe this work opens up new research directions on top of the newly proposed zero-bit watermark methods.  We do not _claim_ to propose a ready-made solution for embedding multi-bit watermark onto language model outputs. We will revise our paper in the following days to avoid any confusion.
>
> Another point to take into consideration is that from a protection standpoint, enforcing the adversary to take evading measures is itself valuable if it is costly. The best -- arguably -- paraphrasing model, ChatGPT is behind paywall when called via API.
>
> ## Comparison to Steganography
>
> The below response is identical to the general response.
>
> Watermarking also differs from steganography works (Ziegler et al., 2019, de Witt et al., 2022) in this aspect they do not consider possible corruption. In particular, Ziegler et al. rely on sequential arithmetic coding of every token. Thus, substituting or removing the first token of the cover text will alter the entire extract message. We will try to empirically demonstrate this in the response period, should time allow.
>
>
> ## Methodology
>
> **List decoding parameterization** Thank you for the suggestion. While we did consider using a confidence threshold, this was difficult to realize given that the prediction was uncalibrated. We opted for a fixed-sized list in the experiment for simplicity.  Our result regarding list decoding was presented to show the effectiveness of list decoding in identifying the possible candidates, rather than to claim that this is actually achievable without any further means. ). In Figure 6c, the validity of the confidence measure used in list decoding is demonstrated by showing that it is inversely proportional to the error rate.
>
>
> **Practicality of list decoding** In practice, list decoding is especially useful because provenance tracing via watermarking is far from finding an exact solution, but narrowing down the possible leakage points for a more detailed inspection that may costly. This is because the decoded message is not error-free in the real-world and thus, one cannot finalize the decision by watermarking alone.  For instance, when encoding the timestamp list decoding outputs the _range_ of time, for which the practitioners to manually inspect and find suspicious activity. When encoding the exact user ID, list decoding will output a list of IDs with varying characters in certain positions. Then, the practitioners will consider the IDs that actually exist in their systems and further inspect other details such as recent user activities.

---

> ### Author Response · Authors · 2023-11-14
> **Responses to Weaknesses [2]**
>
> **Practicality of list decoding** In practice, list decoding is especially useful because provenance tracing via watermarking is far from finding an exact solution, but narrowing down the possible leakage points for a more detailed inspection that may be costly. This is because the decoded message is not error-free in the real-world and thus, one cannot finalize the decision by watermarking alone.  For instance, when encoding the timestamp list decoding outputs the _range_ of time, for which the practitioners to manually inspect and find suspicious activity. When encoding the exact user ID, list decoding will output a list of IDs with varying characters in certain positions. Then the practitioners will consider the IDs that actually exist in their systems and further inspect other details such as recent user activities.
>
>
> ## Experiments: metrics and comparison to KGW zero-bit watermark*
>
> We will revise the paper to include the true positive rate as well. Below we present TPR at different FPR thresholds shown in Table 6. This uses 500 positive (watermarked) samples and 1000 negative samples at 250 tokens.
>
> | Bit-width | @FPR=1e-3 |
> |--------| ------|
> |0 | .983|
> |8 | .975|
> |16|.950|
> |24|.931|
> |32|.907|
>
> As bit-width increases, the detection performance decreases. This is expected as the zero-bit setting is a special case of embedding a single-bit message of $\mathbf{m}=$”0” in all the samples. To give an apples-to-apples comparison, we embed the __same__ 8-bit message ($\mathbf{m}$=”00000000”). This is essentially the same setting with the zero-bit watermarking, but applying only Position Allocation on top of it. To see the true positive rate at lower FPR, we sampled 10,000 negative samples and computed the true positive rate. Below, we show the results for zero-bit and 8-bit.
> |Bit-width|@FPR=1e-3 | @FPR=1e-4| @FPR=1e-5|
> |---|--|--|--|
> |0| 0.997|0.997|0.997|
> |8 (single message)| 0.994|0.992|0.992|
> |8 |0.974|0.965|0.965|
>
> As expected, embedding the same message using 8-bit shows nearly identical results. Compared to this, actually embedding the unique 8-bit message does lead to performance degradation, but still maintains a fairly high TPR even at FPR=1e-5.
>
> In Appendix A.8, we discuss in depth the reason behind this is that the detection metric of non-watermarked (human) text is over-estimated as bit-width is increased. In the revised paper, we will mention this limitation. We leave improving the detection methodology for future research.

---

> ### Author Response · Authors · 2023-11-15
> **Response of Questions**
>
> ## Responses to Questions
>
> **Details regarding hash functions**
>
> The hash functions $f$ we used in our experiments are ‘LeftHash’ with $h=1$ and “SelfHash” with $h=4.$
> We chose to experiment on the two hashing schemes based on the previous work presented by Kirchebauer et al. (2023b). Both of the schemes were proposed in Kirchenbauer et al. (2023a). LeftHash is simply dependent on the $h$ token(s) left of the current token. This is implemented as $f(x)= s * \sum_i x_i$   where $s$ is a secret key and $x$ denotes the list of $h$ token ids. SelfHash is a bit more complicated as the output depends on the current vocabulary distribution. Please refer to Algorithm 3 of Kirchenbauer et al. (2023a). The results for SelfHash are in Appendix Table 9 and 10. We give the bit accuracy results across equal bits per token here:
>
> |Bit |4 |8 |16| 32| 64 |
> |--|--|--|--|--|--|
> |LeftHash (h=1)|.961 (.13) |.958 (.09)| .951 (.07)| .913 (.08)| .846 (.09) |
> |SelfHash (h=4)|.952 (.13) |.953 (.10)| .945 (.08)|.911 (.08) |.850 (.08) |
>
> The context width $h$ determines the tradeoff between robustness and privacy (how easy to find out the hash function). Smaller value of $h$ leads to robustness to corruption as the hashing scheme is reliant only on the previous $h$ tokens. On the other hand, this may lead to reverse-engineering of finding out the tokens in the colorlists. Realistically, we believe this is hard even for the easiest case ($h$=1). Assuming the use of the LLaMA-2 tokenizer, the brute-force attack takes $\mathcal{V}^{h+1}= 32000^2 =1e^{9}$, which is a considerable amount of effort. All this presumes that the API call to extracting the watermark is accessible and that the API call shows the total number of colorlisted tokens. Simply displaying the p-value of certain thresholds (e.g. {1e-6, 1e-5, 1e-4, 1e-3, 1e-2}) will make reverse engineering extremely difficult.
>
>
> **In “List Decoding The linear nature with respect to…”, what does “linear nature” refer to?**
>
>
> Thank you for the question!  We believe ‘linear’ is not an accurate depiction. Our intent was to characterize the decoding step as a decomposition of each position, which enables decoding without added latency when the message length is significantly long. This is revised in the paper as
> >The decomposition of the message into each bit position bounds the computation during decoding to the number of tokens. This allows MPAC to output a list [..]
>
>
> **Unclear how scheme works for when $\gamma\neq\frac{1}{r}$**
>
> When $\gamma=0.25$ and $r=2$, we are choosing only from the red and yellow lists (out of the four colored lists) in Figure 1. This case was what motivated us to propose Colorlisting to fully utilize the vocabulary partitions.
>
> **How is the p-value computed in the last paragraph of Page 7 (experiments)? Why is it computed relative to the second runner-up not the null hypothesis?**
>
> The p-value here indicates comparing the two _mean bit accuracies_ of the two best and the runner-up schemes ($\gamma=0.25$ and $r=4, \gamma=0.5, r=2$). The p-value was computed using an unpaired t-test. This is not the p-value of the watermark detection. Sorry for the confusion.
>
>
> **Why $\gamma=0.25, r=4$ outperforms $\gamma=0.5, r=2$? Is this due to the effective bit-width?**
>
> Yes. We believe the benefits of having a shorter message length (less positions to encode) outweigh the disadvantages of choosing from more options for this particular case. Initially, we thought the larger the value of $r$, the better the multibit performance. However, this was not always the case, which led to the discussion in Section 5 and Figure 6b.
>
> What we did find empirically was that for a fixed $\gamma$, larger $r$ benefits multi-bit performance. How $\gamma$ affects the multi-bit performance is a more complicated one, but we did find empirically that $\gamma=0.25, r=4$ outperforms $\gamma=0.5, r=2$ in multi-bit and zero-bit detection. For zero-bit detection, lower $\gamma$ leads to better detection performance, because this reduces the probability of a token from coming from a greenlist for a non-watermarked text.
>
>
> **Where can I find more detail on how the copy-paste attack was carried out?**
>
>
> We followed the copy-paste attack proposed in Kirchenbauer et al., (2023b). A random snippet of the watermarked text is inserted into a human text to comprise $p$ percentage of the entire text. This does not preserve the start and end tokens of the machine text, making it extremely difficult for extracting the watermark that does not rely on sampling the position with the hashing scheme like MPAC. An instance of this watermarking scheme is shown in Figure 2a whereby the position of the message is embedded in a rule-based manner.

---

> ### Author Response · Authors · 2023-11-15
>
> ## Typos
>
> We have fixed the typo. We appreciate the detailed review.
>
> ## Reference
>
> Kirchenbauer, John, et al. "A watermark for large language models." arXiv preprint arXiv:2301.10226 (2023).
>
> Kirchenbauer, John, et al. "On the Reliability of Watermarks for Large Language Models." arXiv preprint arXiv:2306.04634 (2023).

---

### Official Review · Reviewer_eLnz · 2023-10-31

**Soundness:** 3 good
**Presentation:** 3 good
**Contribution:** 3 good
**Rating:** 6
**Confidence:** 2

**Summary:**

The paper presents a novel technique, "Multi-bit Watermark via Position Allocation" (MPAC), to prevent malicious misuses of large language models beyond the identification of machine-generated text. Traditional methods mainly focus on detection, but the current approach embeds traceable multi-bit information during the model's generation. This makes it easier to find the original creator of the item and to notice it, which is important when combating serious misuses like the dissemination of false information on social media. This watermarking method is built upon the previously proposed zero-bit watermarking technique and allows for the embedding of longer messages without the need for fine-tuning the model. The embedded watermark remains robust against manipulations like human text interleaving and paraphrasing. Experiments show that great accuracy can be achieved when embedding messages up to 32 bits. The method bridges the gap between machine and human text distinction while embedding valuable information, offering a viable way for countering large language model abuses.

**Strengths:**

Strength:
•	The introduction gives a concise overview of the significance of machine-generated text identification, the approaches used in the past, and the reasons why more sophisticated techniques are required.
•	The paper provides various graphs and visual representations, aiding in understanding the performance under different conditions.
•	One of the strengths of the paper is its demonstration of the model's performance under copy-paste attacks, which tests the robustness of their watermarking method.

**Weaknesses:**

Weakness:
•	The performance of the model seems to be reliant on token length. When messages are longer, there seems to be a dip in performance, which may not be suitable for all applications. For example: As mentioned under Figure 5, embedding longer messages at fixed bits per token seems to decrease the performance, especially when reaching up to 64-bit.
•	Metrics like AUC and bit-accuracy are essential, and the paper might benefit from a more qualitative or user-centric analysis. Reliance solely on quantitative metrics might not capture the full user experience or real-world applicability. For example: The paper primarily discusses results in terms of AUC, bit-accuracy, and other such metrics. A discussion on real-world applications potential user feedback could provide a more complete picture of the method's usefulness.
•	While the paper evaluates against human-induced copy-paste attacks, it doesn't delve deeply into other potential real-world challenges. For instance, how would the system perform against more sophisticated adversarial attacks or in scenarios with heavy noise?
•	The method's reliability on decoding depends heavily on the pseudo-random generator function. If someone can reverse-engineer or predict the pseudo-random function, the watermark can potentially be tampered with or removed.

**Questions:**

•	While the paper states that the watermark is relatively robust under strong attacks like interleaving human texts and paraphrasing, are there any known methods or strategies that might weaken or remove the watermark effectively?
•	How easily can the MPAC method integrate with existing large language models and platforms? Are there any potential challenges to be aware of?
•	A detailed comparison between MPAC and other watermarking techniques, in terms of performance, accuracy, and robustness, would provide more context to the reader regarding the advantages of MPAC.

---

> ### Author Response · Authors · 2023-11-14
> **Responses to Weaknesses**
>
> Thank you for your insightful feedback and questions. We hope our response addresses your concerns. If you have further concerns that prevent you from raising your score, please let us know.
>
> **W1: Performance reliant on token length**
>
>
> Yes. All watermarking methods rely on the complexity of the medium. This is often quantified as bits per tokens (BPT) or bits per pixel (BPP) in the image domain. Nonetheless, we believe this does not take away our main contribution, which is showing the feasibility of embedding a robust multi-bit watermark onto language model outputs. Please take into consideration that this is one of the first works enabling such an application. The embedding of $\geq$ 32-bit is to show the feasibility of embedding long messages without any latency, which is computationally infeasible on concurrent works  (Fernandez et al., 2023a; Wang et al., 2023).
>
>
> Moreover, we discuss and analyze in depth the limitations of multi-bit watermarking in (1) Section 3.3 and Figure 2b how SNR is a more reliable factor in estimating the performance and (2) the tradeoff between the capacity of the watermark and its detection rate (Section 5 and Appendix A.8.  We believe this discussion is an important contribution in enhancing the performance by the future works.
>
>
> **W2: A discussion on real-world applications potential user feedback could provide a more complete picture**
>
>
> We agree that metrics like AUC does not paint the full picture. To better illustrate the usefulness of the multi-bit watermarking, we give an example scenario of tracing back to the provenance after identifying the machine-text as done in Fernandez et al. (2023). Assume we are embedding 8-bit messages (256 users) onto 250 tokens. Allowing a false positive rate of 1e-3, we can correctly identify 98.3% of the machine text. Among the identified machine text, the exact match of finding the correct user is 92.6%. The details of this scenario is in Appendix A.7.
>
> Another obstacle in deploying watermarking in the real-world is its quality. Luckily, our MPAC maintains the exact quality with zero-bit watermarking (KGW) as shown in Figure 3. The effect of watermarking on the text quality _compared to no watermarking_ shows promising results in human evaluations when sufficiently large models are used for open-ended generation by Kirchenbauer et al. 2023b (Appendix A.2 and A.9).
>
>
> We are happy to discuss different aspects of user experience that the reviewer has in mind!
>
> **W3: While the paper evaluates against human-induced copy-paste attacks, it doesn't delve deeply into other potential real-world challenges. For instance, how would the system perform against more sophisticated adversarial attacks or in scenarios with heavy noise?**
>
>
> We believe our work extensively covers potential attempts to _evade_ watermark extraction by experimenting on paraphrasing on Chat-GPT (Figure 5b) and other language models (Table 4). While we do not believe these are exhaustive, they are easily accessible by potential adversaries and can be automated.
>
> In addition, list decoding is one solution for mitigating real-world challenges. This is especially useful in practice because provenance tracing via watermarking is far from finding an exact solution, but narrowing down the possible leakage points for a more detailed inspection that may be costly. This is because the decoded message is not error-free in the real-world and thus, one cannot finalize the decision by watermarking alone.  For instance, when encoding the timestamp list decoding outputs the _range_ of time, for which the practitioners to manually inspect and find suspicious activity. When encoding the exact user ID, list decoding will output a list of IDs with varying characters in certain positions. Then the practitioners will consider the IDs that actually exist in their systems and further inspect other details such as recent user activities.
>
> That said, we are not claiming to propose a ready-made solution as this work is at an early stage. However, we do believe considering real-world challenges is important and are happy to discuss other sophisticated attacks or challenges in the real world the reviewer has in mind!

---

> ### Author Response · Authors · 2023-11-14
> **Response to Weaknesses [2]**
>
> **W4: If someone can reverse-engineer or predict the pseudo-random function, the watermark can potentially be tampered with or removed.**
>
>
> Watermark spoofing has been mentioned as a possible attack to “fake” a watermark to harm the reputation of certain models (Sadasivan et al., 2023). More realistically, removing the watermark may be possible if one knows the tokens in the colorlist. An important parameter that tradeoff privacy (difficulty of reverse engineering) and robustness (to corruptions) is the context width $h$. Assuming the use of LLaMA-2 tokenizer, the brute-force attack takes $|\mathcal{V}|^{h+1}= 32000^2 \approx 1e^{9}$, which is a considerable amount of effort even for the easiest case ($h$=1). All this presumes that the API call to extracting the watermark is accessible and that it shows the total number of colorlisted tokens. Simply displaying the extracted multi-bit message and the p-value under certain thresholds, e.g. {1e-6, 1e-5, 1e-4, 1e-3, 1e-2}, will make reverse engineering extremely difficult.

---

> ### Author Response · Authors · 2023-11-14
> **Responses to Questions**
>
> ## Responses to Questions
>
> **Q1: While the paper states that the watermark is relatively robust under strong attacks like interleaving human texts and paraphrasing, are there any known methods or strategies that might weaken or remove the watermark effectively?**
>
>
> A strong corruption method is to use a strong paraphraser. In our experiments, we found that Chat-GPT is a strong candidate in breaking the watermarks (Figure 5b).  An example of paraphrased text is shown below:
> ```
> Watermarked:
> Furthermore, the county has committed to providing Amazon with $20 million in cash grants to help increase the amount of affordable housing in the area and improve bus services. To fund this, the county will implement a new tax on office space valuations. Additionally, Amazon will receive $7 million in electricity rebates over 15 years, with an average annual discount of $60,000. These rebates are set to begin in 2020.
>
> Paraphrased:
> The county also has pledged to provide Amazon with $20 million in cash grants, to increase the amount of affordable housing in the area and to upgrade bus service. The county would receive the money through a new tax on office space valuations.
> In addition, the county plans to provide Amazon with a 15-year payment schedule for about $7 million in electricity rebates. This would give the company an average annual discount of $60,000, according to the agreement. The electricity rebates would start in 2020.
> ```
>
>
>
>
> **Q2: How easily can the MPAC method integrate with existing large language models and platforms? Are there any potential challenges to be aware of?**
>
>
> The simplicity of the method makes it a simple extension of KGW’s zero-bit watermarking. As shown in our main paper this leads to a simple step of sampling the position prior to selecting the token subset. Accordingly, this leads to zero overhead due to increasing the message length as shown in Figure 3. Our MPAC is also theoretically extendable to other zero-bit watermarking methods. The works of Aaronson and Kirchner (2023) and Kuditipudi et al., (2023) use exponential minimum sampling with a pre-defined set of watermark keys. Our Position Allocation scheme can be easily adapted as shown in Appendix A.10. The key to this is finding how to discriminate between $\textbf{m}[p]=0$ and $\textbf{m}[p]=1$. Since exponential minimum sampling favors tokens with large $\mathbf{r}_v$, this can be achieved inverting $\mathbf{r}$ by Eq. 8. We will empirically show the results in the future.
>
>
> **Q3: A detailed comparison between MPAC and other watermarking techniques, in terms of performance, accuracy, and robustness, would provide more context to the reader regarding the advantages of MPAC.**
>
> We agree and we are working on presenting the results in the discussion period. Please note that two concurrent works were uploaded in July 2023 (Fernandez et al., 2023a; Wang et al., 2023). The methodological differences our advantages of allowing embedding of long messages are discussed in the Related Section.
>
> Our current submission discusses (1) the limitation of encoding message sequentially in terms of robustness (Figure 2a), (2) how Position Allocation affects the performance in terms of SNR (Figure 2b) and (3) how colorlisting enhances the performance (Figure 4).
>
>
>
> ## Reference
>
> Kirchenbauer, John, et al. "On the Reliability of Watermarks for Large Language Models." arXiv preprint arXiv:2306.04634 (2023).
>
> Vinu Sankar Sadasivan, Aounon Kumar, Sriram Balasubramanian, Wenxiao Wang, and Soheil Feizi. Can AI-generated text be reliably detected?, 2023. arXiv preprint arxiv:2303. 11156 (2023).
>
> Kuditipudi, Rohith, et al. "Robust distortion-free watermarks for language models." arXiv preprint arXiv:2307.15593 (2023).

---

> > ### Comment · Reviewer_eLnz · 2023-11-22
> > **Thanks for the detailed response!**
> >
> > Thank you for such a detailed response. The responses clarified most of my concerns, especially the general response of comparisons with concurrent work, which was helpful. I will consider these responses and other reviewer feedback before submitting the final rating. Thanks again.

---

### Official Review · Reviewer_TksX · 2023-11-03

**Soundness:** 3 good
**Presentation:** 2 fair
**Contribution:** 3 good
**Rating:** 6
**Confidence:** 4

**Summary:**

This paper presents a novel method titled "Multi-bit Watermark via Position Allocation" (MPAC) for tracing the misuse of large language models beyond just identifying machine-generated text. The method embeds traceable multi-bit information during language model generation, enabling robust extraction of the watermark without any model access, embedding and extraction of long messages without fine-tuning, and maintaining text quality. The paper also demonstrates the robustness of the watermark under strong attacks like interleaving human texts and paraphrasing.

**Strengths:**

(1) The paper proposes an innovative and practical method to address the problem of misuse of large language models, extending beyond mere identification to traceability.

(2) The proposed method, Multi-bit Watermark via Position Allocation (MPAC), allows embedding and extraction of long messages without model access or fine-tuning, which is a significant improvement over existing methods.

(3) The authors also offer detailed empirical findings from their experiments, showing that their method can effectively embed 8-bit messages in short text lengths with over 90% bit accuracy.

**Weaknesses:**

(1) The paper did not provide comparisons to other multi-bit watermarking methods, such as the method presented by Yoo et al. in "Robust multi-bit natural language watermarking through invariant features" Providing comparisons to related work would have strengthened the paper.

(2) The detection of the watermark is an important part. More details on the watermark detection methodology would have been beneficial.

(3) Embedding capacity limited by the text distribution and model. Low entropy distributions like code have an inherently lower capacity.

(4) The author could consider testing on other models besides LLAMA.

**Questions:**

None

---

> ### Author Response · Authors · 2023-11-14
> **Author Responses to the Weaknesses**
>
> Thank you for the encouraging comments and the insightful feedbacks. We hope our response addresses your concerns. If you have further concerns that prevent you from raising your score, please let us know.
>
> ## Response to Weaknesses
> **Comparison with other watermarking methods**
>
> As far as we know, the most robust multi-bit watermarking method among post-processing methods (Abdelnabi & Fritz, 2021, Yang et al., 2022)  is Yoo et al. (2023). However, this work only considers corruption at word-levels such as deletion, insertion, and substitution. Methodologically, one clear limitation is the sentence-level method whereby embedding and extraction are done at each sentence level. As an illustration, consider the example below:
> ```
> Sentence 1 -> extracted message : “000”
> Sentence 2 -> extracted message : “000”
> Sentence 3 -> extracted message : “111”
> Sentence 4 -> extracted message : “111”
> Final extracted message : “000000111111”
> ```
> If any human text were to be added into the first part of the writing (instance of copy-paste attack), the message is irrecoverable as it has no way to identify _when_ the machine text starts or ends. The problem persists when a single sentence in the middle is removed as well. In contrast, the proposed algorithm MPAC need not know this and achieves relative robustness in various kinds of attacks. In addition, ours enable zero-bit detection (machine-text detection) and multi-bit watermark all at the same time.
>
> The concurrent works (Wang et al., 2023; Fernandez et al., 2023), which are detailed above in the general comments and Related Works section of the paper, also rely on the hashing function similar to ours. We expect them to have better robustness than the post-processing methods. We are working on presenting the comparison of the robustness and load capacity of these methods in the following days. Theoretically, our MPAC has an advantage in embedding long messages without any added latency during generation (Figure 3).
>
> **More details on the watermark detection method**
>
> We agree that the paper will benefit from having a more detailed explanation on the detection method. The decoding algorithm has been moved into the main text. We will revise the paper in the following days to use the algorithm section to better explain the detection methodology.
>
> **W3: Embedding capacity is limited by text distribution and model & W4: Other models besides LLaMA**
>
> We agree that some of the text distributions inherently have low entropy, preventing them from embedding large, if any, amount of information. Nonetheless, malicious use cases of language model also involves open-ended generation rather than a fixed solution. Generation of disinformation falls into this case as numerous different passages can be created given a certain topic or propaganda. In our main experiments, we resemble this scenario by completing a news-like subset of C4. As shown in our experiments, we can extract a 8-bit message into 125 tokens with $\approx$ 95%.
>
> For reference, the above paragraph contains 123 tokens.
>
> Below we present the results for instruction-tuned models for LLaMA-2-7b (`meta-llama/Llama-2-7b-chat-hf`), Mistral-7b (`mistralai/Mistral-7B-Instruct-v0.1`), and a relatively small model (`OPT-1.3b`) .
>
> The results are on 8-bit and $\approx$ 250 tokens for 100 samples. For the tuned models, the instruction was “Complete the following news article:”. While the finetuned version on LLaMA shows noticeable falloff, this is _not_ the case for Mistral (Jiang et al., 2023).
>
> | Model | Bit Acc. |
> | ---------| ---------|
> | LlaMA-2-7b | .986 (.06) |
> | + chat | .922 (.13) |
> | Mistral-7b | .987 (.06)|
> | +chat | .977 (.08) |
> |OPT-1.3b|.982 (.07)|
>
> This hints at the possibility that instruction-tuning itself does not limit the capacity of watermarking, but rather the dataset and amount of tuning may be the key. We leave further analysis as a future work.
>
>
>
> ## Reference
> Sahar Abdelnabi and Mario Fritz. Adversarial watermarking transformer: Towards tracing text provenance with data hiding. In 2021 IEEE Symposium on Security and Privacy (SP), pp. 121– 140. IEEE, 2021.
>
> Xi Yang, Jie Zhang, Kejiang Chen, Weiming Zhang, Zehua Ma, Feng Wang, and Nenghai Yu. Trac- ing text provenance via context-aware lexical substitution. In Proceedings of the AAAI Conference on Artificial Intelligence, volume 36, pp. 11613–11621, 2022.
>
>
> Yoo, KiYoon, et al. "Robust multi-bit natural language watermarking through invariant features." Proceedings of the 61st Annual Meeting of the Association for Computational Linguistics (Volume 1: Long Papers). 2023.

---

### Official Review · Reviewer_7Lgp · 2023-11-09

**Soundness:** 3 good
**Presentation:** 2 fair
**Contribution:** 2 fair
**Rating:** 5
**Confidence:** 3

**Summary:**

The paper extends the  Kirchenbauer et al. watermarking scheme to encode multi-bit messages. These allow for finer tracking of AI generated content at a more granular level.

The paper does this the use color lists and positional allocation. Kirchenbauer et al. partitions the vocab into 2 lists, but here the authors propose partitioning it into multiple color lists, and picking the color for sampling based on the information in the message at a certain position. The position in the message is determined by hashing the previously seen context, and then the color in the message at this position is looked up. The authors run various experiments to look at the quality of responses, detectability and robustness of the watermark, and present evidence illustrating the challenges of encoding longer messages.

**Strengths:**

1. Encoding multi-bit information is an important problem, and a useful extension to 0-bit watermarking. This lets us specific identify users abusing LLMs, going beyond the problem of simply detecting LLM generated content.

2. The approach presented seems reasonable, and an improvement over simple baselines of: 1) a unique watermarking key for each message we want to encode,  or 2) r=2 (the Kirchenbauer set up) and then simply picking the red and green lists based on the position.

3. The experiments are thorough, and the method seems to work for encoding multi-bit messages without degrading the quality anymore than the 0-bit scheme (the 0-bit scheme already degrades the LLM quality a bit).

**Weaknesses:**

1. I found the presentation a bit difficult to follow. I think the paper would be much easier to follow if there were fewer forward references, and the paper was a bit more self contained. It might also be worth dedicating space to central contributions in the paper, and having a list of claimed contributions in the paper.

I would like to see the decoding scheme in the main paper, and not the appendix since that is central to understanding a lot of the other details that are in the main paper.

2. There are concerns that there's not enough bandwidth to even do 0-bit watermarking, particularly for LLMs that have gone through RLHF/SFT as that process significantly reduces the entropy available for watermarking. It is unclear to me if the proposed approach is useful in practice for encoding multi-bit messages.

3. There are a few details missing. It is claimed that list decoding improves performance, but it is unclear to me how the decoded list is used in this accuracy computation. If one of the messages in the list matches the encoded message, is that counted as a success?

4. I think the innovation over the 0-bit UMD scheme is somewhat incremental for a machine learning conference, at least from an ML standpoint.

**Questions:**

1.  The UMD scheme is inherently distortionary. Is there a way to apply the ideas presented in this paper to the non-distortionary schemes (e.g., https://arxiv.org/abs/2307.15593)?

2. Can you measure the performance of the multi-bit encoding scheme on IT/RLHF/SFT models like Llama or Vicuna? They have much lesser entropy and are closer to the models deployed in the real world. It would be interesting to see if there is bandwidth for multi-bit watermarking for those models.

3. How is the accuracy computed with list decoding?

4. Can you show some ablation experiments depicting the comparisons between list decoding, and random bit flips in the most likely decoded message?

5. The Kirchenbauer et al. method degrades the language model with increasing delta. Would it be possible to compare the degradation with changing delta for their 0-bit scheme, and the multi-bit scheme presented here?

---

> ### Author Response · Authors · 2023-11-14
> **Response regarding weaknesses**
>
> Thank you for the feedback and the interesting questions. Below, we have responded to your concerns and questions. We hope our response addresses your concerns. If you have further questions that prevent you from raising your score, please let us know.
>
> ## Responses to Weaknesses
>
> **Regarding presentation and the decoding scheme**
>
> Thank you for the suggestions. We also believe this would improve the readability for the general readers and will reflected in the revised paper. Our decoding algorithm is now located at the end of Section 3.2.
>
> To summarize our key contributions,
>
> 1.	We propose and validate Position Allocation that extends zero-bit watermark to multi-bit. Additionally, we show Colorlisting can further enhance multi-bit performance.
> 2.	Through this, we demonstrate the feasibility of directly embedding long multi-bit information into language model outputs. This allows a robust watermark that can withstand realistic attacks that alter the start and end positions of the machine text (copy-paste attack) and change the entire sentence structures (paraphrasing) unlike previous multi-bit watermark works.
> 3.	We discuss and analyze the limitations of multi-bit watermarking – namely, the tradeoff between the capacity of the watermark and its detection rate (Section 5 and Appendix A.8.
>
> We will be incorporating the contributions in the paper in the future.
>
>
> **“Not enough bandwidth for multi-bit message” and Results for instruction-tuned models**
>
> Certain tasks such as code generation have shown to be difficult for watermarking due to the inherently low entropy distribution. Nonetheless, many other open-ended generation tasks certainly have enough capacity for multi-bit watermark. Generation of disinformation falls into this case as numerous different passages can be created given a certain topic or propaganda. Our main experiments simulate this scenario by completing a news-like subset of C4. As shown in our experiments, we can extract a 8-bit message into 125 tokens with $\approx$ 95%.
>
> For reference, the above paragraph contains 123 tokens.
>
> Below we present the results for instruction-tuned models for LLaMA-2-7b (`meta-llama/Llama-2-7b-chat-hf`) and Mistral-7b (`mistralai/Mistral-7B-Instruct-v0.1`). The results are on 8-bit and $\approx$ 250 tokens for 100 samples. For the tuned models, the instruction was “Complete the following news article:”. While the finetuned version on LLaMA shows noticeable falloff, this is _not_ the case for Mistral (Jiang et al., 2023).
>
> | Model | Bit Acc. |
> | ---------| ---------|
> | LlaMA-2-7b | .986 (.06) |
> | + chat | .922 (.13) |
> | Mistral-7b | .987 (.06)|
> | +chat | .977 (.08) |
>
>  This hints at the possibility that instruction-tuning itself does not limit the capacity of watermarking, but rather the dataset and amount of tuning may be the key. We leave further analysis as a future work.
>
> **Details on list decoding**
>
> For computing the bit accuracy of list decoding, we take the “closest message out of the candidates” (Section 4.1 Metrics), which is assuming the best-case scenario. This was presented to show the effectiveness of list decoding in identifying the possible candidates, rather than to claim that this is actually achievable without any further means. In addition, we also note in the end of Section 3.3 that
> >list decoding technique is not unique to ours and can be applied to other methods as long as the decoding stage is computationally feasible.
>
> Nonetheless, the ability to encode long messages without any added latency is unique to our algorithm due to Position Allocation (compared to the concurrent works). In Figure 6c, the validity of the confidence measure used in list decoding is demonstrated by showing that it is inversely proportional to the error rate.
>
> In practice, list decoding is especially useful because provenance tracing via watermarking is far from finding an exact solution, but narrowing down the possible leakage points for a more detailed inspection that may costly. This is because the decoded message is not error-free in the real-world and thus, one cannot finalize the decision by watermarking alone.  For instance, when encoding the timestamp list decoding outputs the _range_ of time, for which the practitioners to manually inspect and find suspicious activity. When encoding the exact user ID, list decoding will output a list of IDs with varying characters in certain positions. Then the practitioners will consider the IDs that actually exist in their systems and further inspect other details such as recent user activities.

---

> ### Author Response · Authors · 2023-11-14
> **Responses to Questions**
>
> ## Response to Questions
>
> **Q1: The UMD scheme is inherently distortionary. Is there a way to apply the ideas presented in this paper to the non-distortionary schemes?**
>
> The cited work is similar to Aaronson and Kirchner, 2023 in that it uses exponential minimum sampling with a pre-defined set of watermark keys. Our Position Allocation scheme can be easily adapted as shown in Appendix A.10. The key to this is finding how to discriminate between $\textbf{m}[p]=0$ and $\textbf{m}[p]=1$. Since exponential minimum sampling favors tokens with large $\mathbf{r}_v$, this can be achieved inverting $\mathbf{r}$ by Eq. 8. We will empirically experiment this in the future.
>
>
> **Q3: How is the accuracy computed with list decoding?**
>
> Please find our response above in the comment regarding weakness.
>
> **Q4: Can you show some ablation experiments depicting the comparisons between list decoding, and random bit flips in the most likely decoded message?**
>
> We observed “that randomly altering a single position provides a good list as the best candidate message is already a good starting point” (Section 5). This was likely due to the computing 16-lists when the effective bit-width was  $\leq$ 16. We will try to provide some numbers in the following days. The comparison with an entirely random list is shown in Table 3.
>
> **Q5: Would it be possible to compare the degradation with changing delta for their 0-bit scheme, and the multi-bit scheme presented here?**
>
> The table below (also in Figure 9) shows the quality and the bit accuracy tradeoff when increasing $\delta $ for embedding 8-bit message into 100 token sequences. P-SP is the metric we used in our main paper that computes the similarity with the watermarked sentences with the human-written sentences.  While we do not have results for the zero-bit watermark, the quality will be equivalent as shown in Figure 3, regardless of the magnitude of $\delta$.
>
>
> |Delta | 0.5 | 1 | 2 | 3| 5 | 10 |
> |-------| -----|---|---|--|---|-----|
> |Bit Acc.| 0.618 (.18) |	0.751 (.19)|0.917 (.13)|0.979 (.08)|0.997 (.02)|0.994 (.04)|
> |P-SP|0.384|0.385|0.376|0.356|0.325|0.244|
>
> ## References
>
> Jiang, Albert Q., et al. "Mistral 7B." arXiv preprint arXiv:2310.06825 (2023).

---

> > ### Author Response · Authors · 2023-11-15
> > **Update on random list decoding**
> >
> > Please find the results of random list decoding below. This was tested on $b=32$, $r=4$ so the effective message length is 16.
> > We show the absolute accuracy improvement over the best message 87.0%:
> > |List Size:|2|4|8|16|
> > |-|-|-|-|-|
> > |Random flip|0.001|0.002|0.002|0.003|
> > |Random flip from best| 0.012|0.020|0.031|0.043|
> > |List decoding by confidence|0.025|0.035|0.047|0.057|
> >
> > As noted in the main text, applying random flips from the best message already provides a good list.
> > Choosing the positions by confidence consistently improves over the random flip.

---

### Author Response · Authors · 2023-11-11
**First author response: major points [1]**

We sincerely thank the reviewers for taking the time to review our work. We would like to provide a preliminary response regarding some points that were commonly raised by the reviewers. We will try to respond to all the concerns raised by the reviewers and clarify any misunderstandings during the response period. The source code is released here: https://github.com/anoymous92874838/multibit-watermark-for-llms.

We are glad most of the reviewers recognized the importance of multi-bit watermarking in language models as “__innovative__” (`TksX`, `Fyz9`) and “__important__” (`7Lgp`) in this research domain. The reviewers also found our “__experiments […]  thorough__” (`7Lgp`, `hfyX`) and “__detailed__” (`TksX`), especially considering the “__demonstration of the model's performance under copy-paste attacks__” (`eLnz`) as one of the strengths.

## Our contributions
To recap, our work is one of the first works directly watermarking language model outputs in a multi-bit manner along with the two concurrent works uploaded in July 2023 (Fernandez et al., 2023a; Wang et al., 2023), which are discussed in the last paragraph of Section 2. Our Position Allocation scheme, together with Colorlisting, demonstrates the feasibility of embedding a multi-bit watermark to counteract high-stake misuses. We do _not_ claim to propose a ready-made solution for embedding multi-bit watermark onto language model outputs. Nonetheless, we believe the generalizability of our framework allows easily extending the multi-bit embedding techniques to other zero-bit watermark methods that favor a subset of the vocabulary (as noted by Reviewer `hfyX` as one of our strengths). An example of applying our method to another watermarking scheme (Aaronson & Kirchner, 2023) is shown Appendix A.10.

## Concerns regarding practical usage
There were some concerns regarding the practicality of multi-bit watermarking as the embedding capacity is limited in some text distributions and models and relies on the lengths of the tokens (`7Lgp`, `TksX`, `eLnz`, ` hfyX`). We agree that this is the case as already demonstrated in many prior works in zero-bit watermarking. Since MPAC also builds upon the zero-bit watermarking framework, this also applies to us.
However, we would like to note that our work is one of the first works that demonstrates the feasibility of embedding a robust multi-bit watermark. In fact, watermarking language model outputs in a zero-bit manner (Kirchenbauer et al., 2023) was proposed less than a year ago. Subsequently, many techniques have been studied on top of it to enhance it in low entropy distributions by using thresholds (Lee et al., 2023) and auxiliary language models (Wang et al., 2023). As noted above, our Position Allocation and Colorlisting schemes are easily generalizable to methods based on favoring token subsets. This means that the benefits of such techniques can be carried onto our proposed method as well as noted by Reviewer `hfyX`.

_Impossible to watermark open-source LLMs_ (`hfyX`)

Yes, this is true. Even so, we believe studying watermarking methods for close-sourced language models is important because of its prevalence. The largest API provider, ChatGPT, had over 206 million monthly unique visitors in April, while Bard had over 140 million monthly unique visitors in May. Moreover, companies in Korea and China have also launched servicing their close-sourced models as API or apps (Baek, 2023;  David, 2023). As of now, the most capable models are arguably close-sourced and if this trend were to continue, it is likely that the number of users will not decline. Watermarking is one of the solutions that can mitigate malicious uses of closed-sourced language models. Having a reference point in academia strongly incentivizes the owners of closed-sourced models to apply them.

---

> ### Author Response · Authors · 2023-11-11
> **First author response: major points [2]**
>
> ## Where our work stands
> “_it is essential for the authors to demonstrate superior robustness [over Kirchenbauer et al., 2023] to substantiate their contributions effectively_” (`Fyz9`)
>
> We respectfully disagree with the premises of Reviewer `Fyz9`. Our work does not claim to “_enhance the robustness intrinsic to_” Kirchenbauer et al., 2023 (KGW hereafter). Rather, we propose a new research direction for extending this to multi-bit. Comparing the performance directly with KGW is not an apples-to-apples comparison. In Section 3.2 Message Encoding, we showed that the zero-bit detection problem is equivalent to embedding a single-bit. As watermarking (Vleeschouwer et al., 2002) involves a tradeoff between load capacity (how much information) and robustness (how much information is preserved), we tackle an inherently more challenging task. Nonetheless, we believe providing a comparison does make the work more complete. As such, we discuss in detail the tradeoff between message length and detection in Section 5 and present FPR comparison with zero-bit detection in Table 6.
>
> In addition, we do not “_purport that the multi-bit watermark sustains the caliber of the text_” compared to non-watermarked text. As stated in our introduction and Figure 3, we state that our text quality is maintained compared to KGW. We will revise the wording to avoid any further confusion.
>
> *Comparison with steganography* (`hfyX`)
>
> Watermarking differs from steganography works (Ziegler et al., 2019, de Witt et al., 2022) as they do not consider possible corruption. For instance, Ziegler et al. rely on sequential arithmetic coding of every token. Thus, substituting or removing the first token of the cover text will alter the entire extract message. We will try to empirically demonstrate this in the response period, should time allow. To the best of our knowledge, no work in both (multi-bit) watermarking and steganography consider robustness as extensively as ours. We experiment on realistic and “difficult” corruptions that interleave machine texts with human texts (copy-paste), paraphrasing based on GPT-3.5 and DIPPER, a paraphrasing model developed specifically for evading machine-text detection (Krishna et al., 2023).
>
>
> Once again, we thank the reviewers for their service. We will update a more detailed response specific to each reviewer’s concern in the individual comments.
>
>
>
> - Scott Aaronson and Hendrik Kirchner. Watermarking gpt outputs. https://www.
> scottaaronson.com/talks/watermark.ppt, 2023. Accessed: 2023-09-14.
> - Sahar Abdelnabi and Mario Fritz. Adversarial watermarking transformer: Towards tracing text provenance with data hiding. In 2021 IEEE Symposium on Security and Privacy (SP), pp. 121– 140. IEEE, 2021.
> - Pierre Fernandez, Antoine Chaffin, Karim Tit, Vivien Chappelier, and Teddy Furon. Three bricks to consolidate watermarks for large language models. arXiv preprint arXiv:2308.00113, 2023a.
> - John Kirchenbauer, Jonas Geiping, Yuxin Wen, Jonathan Katz, Ian Miers, and Tom Goldstein. A watermark for large language models. arXiv preprint arXiv:2301.10226, 2023a.
> - Taehyun Lee, Seokhee Hong, Jaewoo Ahn, Ilgee Hong, Hwaran Lee, Sangdoo Yun, Jamin Shin, and Gunhee Kim. Who wrote this code? watermarking for code generation. arXiv preprint arXiv:2305.15060, 2023.
> - Lean Wang, Wenkai Yang, Deli Chen, Hao Zhou, Yankai Lin, Fandong Meng, Jie Zhou, and Xu Sun. Towards codable text watermarking for large language models. arXiv preprint arXiv:2307.15992, 2023a.
> - KiYoon Yoo, Wonhyuk Ahn, Jiho Jang, and Nojun Kwak. Robust multi-bit natural language water- marking through invariant features. In Proceedings of the 61st Annual Meeting of the Association for Computational Linguistics (Volume 1: Long Papers), pp. 2092–2115, 2023.
> - C. De Vleeschouwer, JF Delaigle and B. Macq, "Invisibility and application functionalities in perceptual watermarking an overview," in Proceedings of the IEEE, vol. 90, no. 1, pp. 64-77, Jan. 2002, doi: 10.1109/5.982406.
>  - Ziegler, Zachary, Yuntian Deng, and Alexander M. Rush. "Neural Linguistic Steganography." Proceedings of the 2019 Conference on Empirical Methods in Natural Language Processing and the 9th International Joint Conference on Natural Language Processing (EMNLP-IJCNLP). 2019.
> - de Witt, Christian Schroeder, et al. "Perfectly secure steganography using minimum entropy coupling." arXiv preprint arXiv:2210.14889 (2022).
> - David F. Carr, As ChatGPT Growth Flattened in May, Google Bard Rose 187%, https://www.similarweb.com/blog/insights/ai-news/chatgpt-bard/, Accessed: 2023-11-10
> - Byung-yeul Baek, Naver to launch HyperCLOVA X AI service to rival ChatGPT, https://www.koreatimes.co.kr/www/tech/2023/11/129_350568.html, Accessed: 2023-11-10
> - Emillia David, Baidu launches Ernie chatbot after Chinese government approval, https://www.theverge.com/2023/8/31/23853878/baidu-launch-ernie-ai-chatbot-china,  Accessed: 2023-11-10

---

### Author Response · Authors · 2023-11-17
**General response**

Dear reviewers,

We have revised our paper based on the feedbacks. Here is a summary of the revisions:
- The decoding algorithm (Algo. 1) has been relocated to the end of Section 3.2.
- For clarity in the watermark detection methodology, we added reference to Algo. 1 (page 7).
-	We added some explanation regarding the differences with steganography in page 2.
-	We added some numerical results of TPR in Appendix to the main paper (page 9). We also mention the discussion of this in the introduction (page 2).
- ~We revised the wording in the abstract to avoid sounding that our work maintains the text with non-watermarked text.~ Edit: Since the abstract cannot be revised during the response period according to the guideline, we rolled back to the original abstract.


The experimental results (TPR at lower FPR thresholds, comparison to other methods; see below) provided in the response period will be reflected in the future version as well.

We would appreciate it greatly if you could re-evaluate our work based on the revision and the responses to check whether we have addressed your comments in a satisfactory manner. We would welcome the opportunity to discuss any remaining concerns and questions that you may have.

---

## Comparison with concurrent work

We ran some experiments comparing our work with Fernandez et al., (2023), which relies on a pseudo-random function similar to ours. **_In summary, the clean performance and robust performance are similar in low bit-widths, but MPAC starts to outperform at 16-bit_**.

Their work uses message-specific secret keys to embed the watermark. For the zero-bit framework of Kirchenbauer et al., (2023), they cyclically shift the greenlist `m` times. This provides `m` distinct signals unique for each message. Since the size of the greenlist is equal to vocabulary size $|\mathcal{V}|$, the bit-width is bounded by $log_2 |\mathcal{V}|$. We dub this framework as ‘greenlist’.

For the zero-bit framework of Aaronson et al. (2023), a random secret key of arbitrary size can be created and cyclically shifted `m` times. This allows embedding a larger bit-width than the aforementioned methods. We dub this framework as ‘exponential minimum sampling’ (EMS).

We compare with both frameworks. For extending the former framework, using the Llama-2-7B tokenizer bounds the bit-width roughly to 15 bits. The results are on 250 tokens and the copy-paste percentage denotes the percentage of non-watermarked texts (higher denotes stronger attack). We report the mean bit accuracy of roughly 500 samples.

For 8-bit, all methods show high performance for the uncorrupted input. Although the gaps are small, Fernandez et al.-greenlist consistently shows the highest performance. Fernandez et al.-EMS shows a considerable drop-off when the attack percentage is high.

|Method|Clean|cp=10%|cp=30%|cp=50%|
|--|--|--|--|--|
|Ours|.986 (.06)|.981 (.07)|.956 (.10)|.900 (.13)|
| Fernandez et al.-EMS |.979 (.10)|.943 (.17)|.858 (.24)|.800 (.28)|
| Fernandez et al.-greenlist |.995 (.05)|.988 (.08)|.970 (.12)|.908 (.20)|


For 16-bit, Fernandez et al.-EMS shows a considerable drop-off. For reference we show Fernandez et al.-greenlist with payload of 32,000 (roughly 15-bit).

|Method|Clean|cp=10%|cp=30%|cp=50%|
|--|--|--|--|--|
|Ours|.951 (.07)|.939 (.08)|.887 (.09)|.819 (.12)|
| Fernandez et al.-EMS |.905 (.20)|.811 (.26)|.702 (.26)|.601 (.23)|
| Fernandez et al.-greenlist (15-bit)|.986 (.09)|.974 (.12)|.929 (.18)|.765 (.26)|

As mentioned in the paper, Fernandez et al. (2023) have a computation cost that exponentially scales with the bit-width. To test this, we ran Fernandez et al.-EMS to embed 24-bit messages for a few samples. Comparing the wall clock time of embedding 16-bit and 24-bit messages, the wall clock time during encoding (generation) is increased by roughly 3.5x (14 seconds $\rightarrow$ 50 seconds), which affects the API user. That of the decoding (message extraction) is increased by 130x (0.87 second $\rightarrow$ 115 seconds per sample), which affects the watermark extractor. This trend showcases embedding longer messages will likely be extremely costly. In contrast, our latency is constant regardless of the bit-width (Fig. 3).

---

### Author Response · Authors · 2023-11-21
**Would greatly appreciate your response**

Dear all reviewers,

This will be our final response. We would greatly appreciate your response.
We hope you find our updated paper and consider re-evaluating our paper, if your concerns have been addressed.

We have updated our experiments section to include comparisons with the two concurrent works. To summarize the results, we show that our method is on par with them and even outperform them in higher bit-width ($b=16$) and higher corruption setting. The increase in bit-width is not bounded unlike Fernandez et al.-greenlist and comes with no latency cost.

This has been reflected in the main experiments.  We believe these results strengthen the paper greatly.

**8b**
|Method|Clean|cp=10%|30%|50%|
|--|--|--|--|--|
|Ours|.986 (.06)|.981 (.07)|.956 (.10)|.900 (.13)|
| Fernandez et al.-greenlist |.995 (.05)|.988 (.08)|.970 (.12)|.908 (.20)|
| Fernandez et al.-EMS |.979 (.10)|.943 (.17)|.858 (.24)|.800 (.28)|
| Wang et al. | .977 (.11) |	.973 (.12)|.951 (.16)|	.858 (.24)|

**16b**
|Method|Clean|cp=10%|30%|50%|
|--|--|--|--|--|
|Ours|.951 (.07)|.939 (.08)|.887 (.09)|.819 (.12)|
| Fernandez et al.-EMS |.905 (.20)|.811 (.26)|.702 (.26)|.601 (.23)|
| Wang et al. |  .936 (.18)	|.909 (.20)	|.810 (.26)	|.614 (.22)|
| Fernandez et al.-greenlist (15-bit)|.986 (.09)|.974 (.12)|.929 (.18)|.765 (.26)|



We thank the reviewers for their time and effort. We found the comments invaluable in improving the paper.

Best,

Authors